# Generalization Principles for Inference over Text-Attributed Graphs with Large Language Models

**Haoyu Wang** [1]  **Shikun Liu** [1]  **Rongzhe Wei** [1]  **Pan Li** [1]

## Abstract

Large language models (LLMs) have recently been introduced to graph learning, aiming to extend their zero-shot generalization success to tasks where labeled graph data is scarce. Among these applications, inference over text-attributed graphs (TAGs) presents unique challenges: existing methods struggle with LLMs' limited context length for processing large node neighborhoods and the misalignment between node embeddings and the LLM token space. To address these issues, we establish two key principles for ensuring generalization and derive the framework LLM-BP accordingly: (1) **Unifying the attribute space with task-adaptive embeddings**, where we leverage LLM-based encoders and task-aware prompting to enhance generalization of the text attribute embeddings; (2) **Developing a generalizable graph information aggregation mechanism**, for which we adopt belief propagation with LLM-estimated parameters that adapt across graphs. Evaluations on 11 real-world TAG benchmarks demonstrate that LLM-BP significantly outperforms existing approaches, achieving 8.10% improvement with task-conditional embeddings and an additional 1.71% gain from adaptive aggregation. The code[2] and task-adaptive embeddings[3] are publicly available.

.

## 1. Introduction

Inspired by the remarkable generalization capabilities of foundation models for text and image data (Achiam et al., 2023; Liu et al., 2021; Radford et al., 2021), researchers have recently explored extending these successes to graph data (Liu et al., 2023b; Mao et al., 2024; Zhao et al., 2023a; Fan et al., 2024; He et al., 2024), aiming to develop models that generalize to new or unseen graphs and thereby reduce reliance on costly human annotation (Li et al., 2024f; Chen et al., 2024d; Feng et al., 2024; Li et al., 2024g). Among various types of graph data, *text-attributed graphs* (TAGs) have found a wide range of applications. These graphs combine both topological relationships and textual attributes associated with each node, which naturally arises in recommendation systems (where user and item nodes may have textual descriptions or reviews) (Bobadilla et al., 2013), academic graphs (where publications include extensive textual metadata) (McCallum et al., 2000; Giles et al., 1998), and financial networks (where transactions and accounts come with textual records) (Kumar et al., 2016; 2018). Given the labeling challenges posed by cold-start problems in recommendation systems or fraud detection in financial networks, methods that can operate with limited labeled data are crucial. In particular, robust zero-shot node labeling across unseen TAGs has become an area of great interest.

Numerous studies have been dedicated to inference tasks on TAGs. Early efforts have primarily focused on adapting pre-trained language model (LM) encoders (Li et al., 2024e; Fang et al., 2024), sometimes in combination with graph neural networks (GNNs) (Hou et al., 2022; Veličković et al., 2018), to incorporate structural information. However, these approaches often struggle to achieve strong generalization performance, largely due to the limited capacity of the underlying models. With the advent of large language models (LLMs) (Kaplan et al., 2020; Huang & Chang, 2022), researchers have proposed two main strategies for integrating LLMs into TAG inference: 1) **Direct Node-Text Input.** Here, raw node texts are directly fed into LLMs. This method demonstrates reasonably good zero-shot performance on TAGs when text attributes are highly informative for node labels (Chen et al., 2024c; Li et al., 2024d). However, when the textual attributes are in-

---

[1]Department of Electrical and Computer Engineering, Georgia Institute of Technology, Atlanta, GA, USA. Correspondence to: Haoyu Wang <haoyu.wang@gatech.edu>, Pan Li <panli@gatech.edu>.

*Proceedings of the 42$^{nd}$ International Conference on Machine Learning*, Vancouver, Canada. PMLR 267, 2025. Copyright 2025 by the author(s).

[2]https://github.com/Graph-COM/LLM_BP
[3]https://huggingface.co/datasets/Graph-COM/Text-Attributed-Graphs

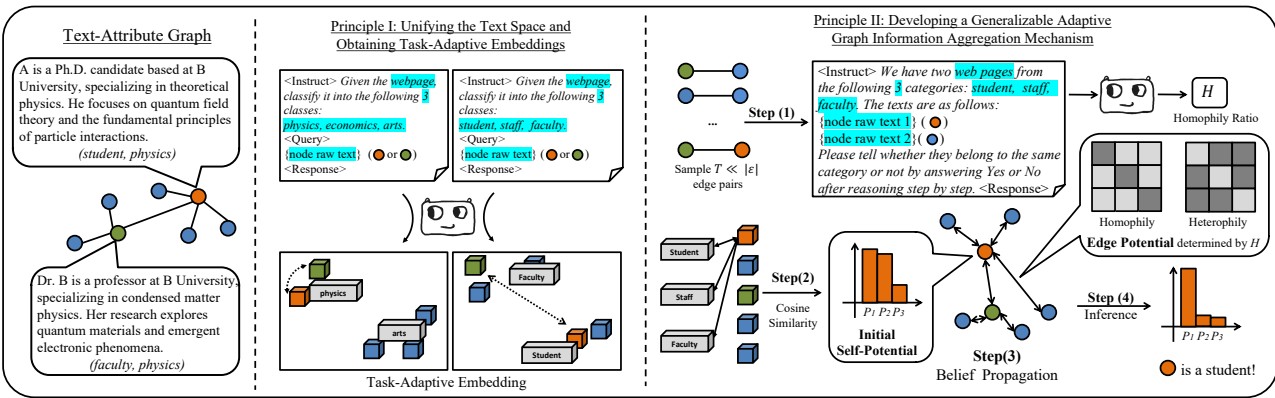

Figure 1: The two generalization principles and the framework of LLM-BP.

sufficient, it becomes necessary to aggregate information from a larger neighborhood in the graph, while this is constrained by the limited context length LLMs can digest and reason over. 2) **Embedding-Based Integration.** In this approach, node texts and their neighboring structural information are first encoded into compressed embeddings, which are then processed by LLMs (Chen et al., 2024b; Tang et al., 2024a; Luo et al., 2024; Wang et al., 2024; Zhang et al., 2024a). Because LLMs are not inherently trained on arbitrary embedding spaces, aligning these embeddings with the LLM's token space is essential - an idea partly inspired by how vision-language models align multimodal data (Radford et al., 2021; Zhai et al., 2023). However, unlike the vision-language domain, where large-scale text–image pairs (Schuhmann et al., 2022) are abundant, the graph domain typically lacks comparable datasets. This scarcity reduces the model's generalization in practice.

In contrast to prior heuristic approaches, this work aims to design a method from first principles for robust zeroshot generalization on TAGs. Because TAGs are inherently multimodal, the proposed method must simultaneously address potential distribution shifts in both textual attributes and graph structure. Specifically, text attributes can vary widely, for example from scientific papers (McCallum et al., 2000) to e-commerce product reviews (Ni et al., 2019). Edge connection patterns can range from homophilic graphs such as citation networks, where papers on similar themes are linked (Giles et al., 1998), to heterophilic graphs such as webpages, which connect nodes with distinct topics (Mernyei & Cangea, 2020). Moreover, the labeling task itself can shift which requires a task-adaptive approach to process both textual features and network structure. Consequently, the core insight behind our model design is grounded in the following two key principles.

**Principle I: Unifying the text space and obtaining task-adaptive embeddings.** LLMs offer powerful textunderstanding capabilities that naturally unify the textual feature space, and have recently been shown to benefit from

task-adaptive prompting (Kong et al., 2024). However, to handle the large-scale graph aggregation discussed later, we require these capabilities to extend beyond raw text to an embedding space. Hence, we propose to adopt LLMbased encoder models such as LLM2Vec (BehnamGhader et al., 2024; Li et al., 2024a) for text embedding. Although this approach might appear to be a naive extension of smaller LM-based embedding methods (e.g., those relying on SBERT (Reimers, 2019) or RoBERTa (Liu, 2019)), we argue that leveraging the decoder-induced encoder structure of LLMs is essential for achieving task-adaptive embeddings. In particular, we introduce a novel prompting strategy that encodes text attributes conditioned on inference-task descriptions, enabling significantly improved zero-shot inference - an ability not readily achieved by smaller LM-based embeddings.

**Principle II: Developing a generalizable adaptive graph information aggregation mechanism.** Graph structure determines the node neighboring relationships and thus the information aggregation from which nodes may benefit the inference. Inspired by the belief propagation (BP) algorithm (Murphy et al., 2013) that gives the optimal statistical inference over pairwise correlated random variables, we propose to regard the graph as a Markov Random Field (MRF), each node as a random variable, and mimic BP to aggregate information for node label inference. Because BP is rooted in basic mathematical principles, this approach is widely generalizable. Algorithmic adaptivity across different TAGs hinges on estimating the coupling coefficients in the graphs, which can be done by having LLMs analyze the attributes of sampled pairs of connected nodes. Moreover, this BPinspired approach naturally adapts to varying levels of text attribute quality: nodes with higher-quality text attributes present greater influence on their neighbors, and vice versa.

By applying the two principles outlined above, we propose our new strategy, LLM-BP, for zero-shot inference over TAGs. LLM-BP does not require any training or finetuning. We evaluate LLM-BP on 11 real-world TAG datasets

from various domains, including citation networks (McCallum et al., 2000; Giles et al., 1998; Sen et al., 2008), e-commerce (Ni et al., 2019), knowledge graphs (Mernyei & Cangea, 2020), and webpage networks (Craven et al., 1998), covering both homophilic and heterophilic graph structures.

Experimental results demonstrate the effectiveness of LLM-BP. Notably, our task-conditional embeddings (Principle I) improve performance by $8.10\%$ on average compared to the best LM-based encoders. In addition, our BP-inspired aggregation mechanism (Principle II) provides an extra $1.71\%$ performance gain with our embeddings, demonstrating strong generalization across both homophilic and heterophilic TAGs. Our experiments also reveal that current methods aligning graph-aggregated embeddings to LLM token spaces significantly underperform approaches that simply use smaller LM encoders without even incorporating graph structures. This outcome indicates that the primary source of generalization in these methods is the smaller LM's text embeddings, rather than LLM-based reasoning on embeddings. It reinforces our earlier argument that limited training data hinders effective alignment in this context, urging caution for future work considering this strategy.

## 2. Related Works

Here, we briefly review existing methods by examining how they enable model generalization across TAGs.

**Tuning Smaller LM Encoders.** These methods typically rely on a source-domain graph for training. Notable works include ZeroG (Li et al., 2024e) that tunes SBert (Reimers, 2019) on source datasets to align class-description embeddings with node text, thereby enhancing zero-shot performance on target datasets. Another approach, UniGLM (Fang et al., 2024), fine-tunes BERT (Kenton & Toutanova, 2019) using contrastive learning on source datasets to yield a more generalizable encoder. GNNs trained with UniGLM embeddings in a supervised manner outperform models that directly adopt LM embeddings.

**Training GNNs for Generalization.** These methods focus on leveraging graph structure in a generalizable manner. Among them, graph self-supervised learning (Liu et al., 2022) is particularly common for producing representations without labeled data, often employing contrastive learning or masked modeling (Veličković et al., 2018; Hou et al., 2022; Zhao et al., 2024). GraphMOE is a more recent technique inspired by the success of mixture-of-experts (Shazeer et al., 2017), pre-training parallel graph experts targeting different structures or domains (Hou et al., 2024; Liu et al., 2024; Xia & Huang, 2024; Qin et al., 2023). Others also consider LM-GNN co-training including (He & Hooi, 2024; Zhu et al., 2024) that also follow a constrastive learning idea. Note that, however, all these methods still require training.

In contrast to the above effort that adopts smaller LM encoders, works that involve the use of LLMs are reviewed in the following and may achieve better generalization. More related works including LLM-based data augmentations for GNN training for generalization and LLMs for other graph reasoning tasks can be found in Appendix. A.

**LLMs with Node-Text Input.** LLMs being directly fed with raw node texts demonstrates strong zero-shot ability on TAGs (Chen et al., 2024c; Huang et al., 2024; Li et al., 2024d). However, they suffer from the limitation of not being able to incorporate graph structural information.

**LLMs with Graph-Embedding Input.** With smaller LM-encoded node embeddings, various strategies integrate graph structure by aggregating these embeddings, such as neighborhood-tree traversal or concatenating the averaged embeddings from different hops (Chen et al., 2024b; Tang et al., 2024a; Luo et al., 2024), or via pre-trained GNNs (Zhang et al., 2024a; Wang et al., 2024). As mentioned earlier, these methods rely on aligning embeddings with the LLMs' token space. For instance, LLaGA (Chen et al., 2024b) trains a simple MLP on citation networks and (Wang et al., 2024) employs a linear projector on the ogbn-Arxiv (Hu et al., 2020) dataset, both using the next-token prediction loss, while (Tang et al., 2024a) adopts self-supervised structure-aware graph matching as the training objective. However, due to limited TAG-domain data, the space alignment in these methods often remains undertrained, leading to degraded performance.

**Multi-Task Graph Foundation Models.** More ambitious studies aim to generalize across various graph-related tasks within a single framework. Notable approaches include graph prompting (Liu et al., 2023c), which introduces "prompting nodes" to transform diverse graph tasks into a unified format. These frameworks then train GNNs to address the tasks (Li et al., 2024e; Liu et al., 2023a; 2024) or further integrate LLMs (Yang et al., 2024; Kong et al., 2024). Although these works are impressive, they still fail to achieve zero-shot performance comparable to those methods that focus on specific graph data domains.

## 3. Generalization Principles for LLM-BP

### 3.1. Notations and Problem Formulation

Let $(\mathcal{G}, X, Y)$ represent a TAG of interest, where $\mathcal{G}(\mathcal{V}, \mathcal{E})$ denotes the graph structure, $\mathcal{V}$ is the node set of size $n$, and $\mathcal{E}$ is the edge set. The node textual attributes are represented as $X = \{X_1, ..., X_n\}$, and each node belongs to one of $c$ classes, with labels given by $Y = \{y_1, y_2, ..., y_n\} \in [c]^n$.

The objective is to infer the labels of nodes in TAGs based on the node attributes and graph structure. This study primarily focuses on the **zero-shot** setting, where no labeled data

are assumed to be available in advance. Additionally, a **few-shot** setting is considered, where $k$ labeled nodes are known for each class. Due to space limitations, results for the few-shot setting are provided in Appendix D.6.

## 3.2. Motivation and the Overall Framework

LLMs are commonly used as decoders for next-token prediction. While LLMs excel at processing natural language inputs, they are not inherently compatible with graph data. Recently, some studies have explored methods to integrate graph data into LLMs, primarily for reasoning tasks (Perozzi et al., 2024; Zhang et al., 2024c; Tang et al., 2024b).

In the context of TAGs, accurate node label inference relies on effectively combining the attributes of multiple nodes, especially when a node's individual attributes are insufficient to determine its label. However, as noted earlier, LLMs are constrained by limited context windows, making it challenging to process all attributes from the potentially large set of connected nodes. Traditional approaches to compressing graph structural information involve creating embeddings, such as using GNNs to aggregate information from the target node's neighbors. While effective, these embedding methods do not seamlessly integrate with LLM inputs and often require non-trivial training effort to align the LLMs' token space with the node embedding space (Chen et al., 2024b; Wang et al., 2024; Tang et al., 2024a).

Our approach, LLM-BP, does not confine LLMs to their traditional usage. We first leverage their capabilities to generate task-adaptive node embeddings. Then, instead of requiring LLMs to directly process these embeddings, LLMs are further employed to analyze graph data and provide generalizable guidance in aggregating these embeddings. These two steps are to match the two generalization principles proposed in Sec. 1. Classification is ultimately performed by computing the cosine similarity between the final node embeddings $\mathbf{h}^X = [h_1^X, ..., h_n^X]$ and candidate class embeddings $\mathbf{q}^C = [q_1^C, ..., q_c^C]$. In the zero-shot setting, class embeddings are generated as follows: we randomly sample $l \ll n$ nodes and employ LLMs to infer their labels. The embeddings of sampled nodes form distinct clusters based on LLMs' prediction. We compute the average embedding of the embeddings closest to the cluster center to obtain the class embedding. In the few-shot setting, class embeddings are obtained by averaging the embeddings of labeled nodes within each class. See Appendix. B.2 for details.

## 3.3. Principle I: Task-Adaptive Node Embeddings $\mathbf{h}^X$

Creating generalizable text embeddings is no longer a significant challenge. Even smaller LM encoders, such as SBert (Reimers, 2019), are capable of achieving this. Indeed, most existing works utilize these encoders to generate initial node embeddings for TAGs (Chen et al., 2024d;b;

Tang et al., 2024a; Wang et al., 2024). However, for these embeddings to be directly usable for label prediction without the need for additional transformation models, it is crucial to incorporate task-specific information. In other words, the embeddings must be tailored to the specific task, resulting in what we term task-adaptive embeddings.

Achieving task adaptivity, however, presents a notable challenge. Smaller LM encoders lack the expressive power necessary to encode nuanced task-specific information. This limitation motivates our adoption of LLM-induced encoders, driven by the emergent capabilities of LLMs in contextual learning (Sahoo et al., 2024; Chen et al., 2023).

There have been recent advancements in extending LLMs to generate text embeddings (BehnamGhader et al., 2024; Muennighoff et al., 2022). In our approach, we utilize a form of LLM2Vec (BehnamGhader et al., 2024), which transforms LLM decoders into encoders via retaining the unidirectional attention mechanism. Following the methodology in (Li et al., 2024a), we extract the output embedding of ⟨response⟩ - the token positioned immediately after the inputs - as the text embedding for the input node attributes.

To embed task-specific information into node embeddings, we propose a prompting strategy structured with the following template:

$$\langle\text{Instruct}\rangle\{\text{task\_description}\}\{\text{class\_info}\}\langle\text{query}\rangle X_i\langle\text{response}\rangle. \quad (1)$$

Here, ⟨·⟩ encloses specific tokens. The task details are described in {task_description}, and {class_info} contains the basic information of each class. An example is given in Fig. 1. The class information serves as a crucial contextual enhancement, enabling LLMs to generate embeddings in a conditioned manner. For more detailed on the class-conditional prompt for each dataset used in this study, refer to Appendix. B.2 and E.1.

## 3.4. Principle II: Generalizable Graph Aggregation

Graph structures can provide essential correlated information for node label inference by characterizing the relationships between node attributes and labels (Zhu et al., 2003; Kipf & Welling, 2016; Veličković et al., 2017; Hamilton et al., 2017; Zhu et al., 2020; Wei et al., 2022).

Specifically, we may consider each node's label and attributes as random variables, and each edge as a coupling between them for connected node pairs. The fundamental BP algorithm enables principled statistical inference over this set of correlated random variables (Murphy et al., 2013). Since BP is inherently agnostic to the application domain of the TAG, emulating BP offers a mechanism to aggregate correlation information encoded in the graph structure across domains.

**Markov Random Field Modeling** We consider the joint probability distribution $\mathbb{P}_{\mathcal{G}}(Y, X)$ over the graph where $Y$ and $X$ denotes the random variables of node labels and attributes, respectively. In $\mathbb{P}_{\mathcal{G}}(Y, X)$, the distribution over the node labels given the graph structure is denoted as

$$\mathbb{P}_{\mathcal{G}}(Y) = \frac{1}{Z_{\mathbf{Y}}} \prod_{i \in \mathcal{V}} \phi_i(y_i) \prod_{(i,j) \in \mathcal{E}} \psi_{ij}(y_i, y_j). \quad (2)$$

Here $\phi_i(y_i)$ denotes the unary potential for node $i$, $\psi_{ij}(y_i, y_j)$ is the edge potential capturing the correlation between labels $y_i$ and $y_j$ of adjacent nodes, and $Z_Y$ is the normalization constant. For node attributes, MRF modeling assumes that each node's attributes are conditionally independent of others given the node labels, which can be characterized by the distribution:

$$\mathbb{P}_{\mathcal{G}}(X \mid Y) = \prod_{i \in \mathcal{V}} \mathbb{P}_{\mathcal{G}}(X_i \mid y_i) = \prod_{i \in \mathcal{V}} \varphi_{y_i}(X_i) \quad (3)$$

where $\varphi_{y_i}(X_i)$ captures the likelihood of having node $i$'s attributes $X_i$ given its label $y_i$.

The proposed modeling approach is highly adaptive, as it can capture the varying graph connectivity patterns across different TAGs through interpretable edge potentials. For instance, $\psi_{ij}(y_i, y_j)$ represents the unnormalized likelihood that nodes with labels $y_i$ and $y_j$ are connected. This formulation naturally incorporates the modeling of graph homophily and heterophily: $\psi_{ij}(y_i, y_i) > \psi_{ij}(y_i, y_j)$ indicates homophily, while $\psi_{ij}(y_i, y_i) < \psi_{ij}(y_i, y_j)$ reflects heterophily. Furthermore, $\varphi_{y_i}(X_i)$ enables the model to account for variations in the quality of text attributes (w.r.t. their indicative power for the labels) across different TAGs, further enhancing its adaptivity. For node classification, we can infer $\mathbb{P}_{\mathcal{G}}(Y \mid X) \propto \prod_{i \in \mathcal{V}} \varphi_{X_i}(y_i) \prod_{(i,j) \in \mathcal{E}} \psi_{ij}(y_i, y_j)$ where $\varphi_{X_i}(y_i) = \varphi_{y_i}(X_i) \phi_i(y_i)$.

**Belief Propagation** Exact inference for $\mathbb{P}_{\mathcal{G}}(Y|X)$ is intractable in large-scale graphs with cycles (Koller, 2009). In practice, loopy belief propagation (LBP) is often used to conduct an approximate inference (Murphy et al., 2013), which follows: Initialize the distributions $p_j^{(0)}(y_j) \propto \varphi_{X_i}(y_i)$ and $m_{i \to j}^{(0)}(y_j) = 1/c$ for all $i, j \in \mathcal{V}$. For $k = 1, 2, ..., L$, we do

$$\log m_{j \to i}^{(k)}(y_i) \cong \text{LSE}_{y_j} [\log \psi_{ij}(y_i, y_j) + \quad (4)$$
$$\log p_j^{(k-1)}(y_j) - \log m_{i \to j}^{(k-1)}(y_j)],$$
$$\log p_i^{(k)}(y_i) \cong \log p_i^{(0)}(y_i) + \sum_{j \in \mathcal{N}(i)} \log m_{j \to i}^{(k)}(y_i),$$

where $\cong$ denotes the equality with difference up-to a constant. LSE stands for the log-sum-exp function: $\text{LSE}_{y_j}[f(y_i, y_j)] = \log \left[ \sum_{y_j} \exp(f(y_i, y_j)) \right]$. The final $\arg \max_{y_i} p_i^{(k)}(y_i)$ gives the label prediction. Detailed derivation can be found in Appendix. C.

---

**Algorithm 1** LLM-BP

**input** TAG $(\mathcal{G}, \boldsymbol{X})$
**output** Class label prediction $\{\hat{y}_i\}_{i \in [n]}$
1: $\boldsymbol{h}^X \leftarrow$ Task-adaptive encoding following Eq. (1)
2: **if** zero-shot **then**
3:     Sample $l \ll n$ nodes, infer labels with LLMs,
4:     Nodes clusters based on LLM prediction,
5:     $\mathbf{q}^C \leftarrow$ Average embedding of samples near center,
6: **else if** few-shot **then**
7:     $\mathbf{q}^C \leftarrow$ Average embedding of $k$ samples per class,
8: **end if**
9: Estimate $\psi_{ij}(y_i, y_j)$ by employing the LLM to analyze the graph data (e.g., using Eq. (6) based on the estimated homophily level $r$.)
10: Initialize $p^{(0)}(y_i) \leftarrow$ Eq. (5) and $m_{i \to j}^{(0)}(y_j) = 1$
11: Run LLM-BP (Eq. (4)) for $L$ iterations or its approximation (Eq. (7)) for single iteration
12: $\hat{y}_i \leftarrow \arg \max_{y_i} \log p_i^{(k)}(y_i; x_i)$

---

**LLM-BP** To execute the above LBP algorithm, we need to specify several components based on the TAG. First, $p_i^{(0)}(y_i)$ represents the distribution of the label $y_i$ given the observed attributes $X_i$ alone, which can be estimated using normalized cosine similarities:

$$p_i^{(0)}(y_i) = \text{softmax}(\{\cos(h_i^X, h_k^C)/\tau\}_{k \in [c]}) \quad (5)$$

where $h_i^X$ and $h_k^C$ denote node $i$'s class-conditional embedding and class $k$'s embedding given by the LLM encoder as discussed in the previous section. $\cos(\cdot)$ denotes cosine similarity and $\tau$ is the temperature hyper-parameter.

Second, we characterize the edge potentials $\psi_{ij}(y_i, y_j)$. We employ an LLM agent to assess the homophily level of the TAG. Specifically, we uniformly at random sample $T$ connected node pairs ($T \ll |\mathcal{E}|$), and for each pair, we prompt the LLM to determine whether the two nodes belong to the same class based on their attributes, as illustrated in Fig. 1. The ratio of "Yes" responses, denoted by $r$ is used to set

$$\psi_{ij}(y_i, y_j) = \begin{cases} r, & \text{if } y_i = y_j \\ 1 - r, & \text{if } y_i \neq y_j \end{cases}, \quad (6)$$

Note that a more complex $\psi_{ij}(y_i, y_j)$ can be adopted by estimating the edge probabilities between any two classes. However, we choose the homophily level as a proof of concept. LLMs can provide a reasonably accurate estimation of the homophily level, as pairwise comparisons are typically much simpler tasks compared to full-scale classification.

(Wei et al., 2022) demonstrated that linear propagation can approximate a single iteration of LBP when feature quality

| Dataset | Text Domain | Graph Structure |
|---|---|---|
| Cora (McCallum et al., 2000) | CS Publication | Homopholic |
| Citeseer (Giles et al., 1998) | CS Publication | Homopholic |
| Pubmed (Sen et al., 2008) | Medical Publication | Homopholic |
| History (Ni et al., 2019) | History Books | Homopholic |
| Children (Ni et al., 2019) | Children Literature | Homopholic |
| Sportsfit (Ni et al., 2019) | Sports Goods | Homopholic |
| Wikics (Mernyei & Cangea, 2020) | Knowledge Graph | Homopholic |
| Cornell (Craven et al., 1998) | School Webpage | Heterophilic |
| Texas (Craven et al., 1998) | School Webpage | Heterophilic |
| Wisconsin (Craven et al., 1998) | School Webpage | Heterophilic |
| Washington (Craven et al., 1998) | School Webpage | Heterophilic |

Table 1: TAG Datasets selected in experiments.

is limited. Based on this insight, we adopt the following approximate LBP formulation (denoted as BP appr.):

$$\log p_i^{(1)}(y_i) \cong \log p_i^{(0)}(y_i) + \qquad (7)$$
$$\text{sgn}(\log \frac{r}{1-r}) \sum_{j \in \mathcal{N}(i)} \log p_j^{(0)}(y_i),$$

where the homophily level $r$ influences the sign of the log-likelihood aggregation from neighboring nodes. We summarize the overall pipeline in Algorithm. 1

## 4. Experiments

In this section, we evaluate LLM-BP based on its two design principles, with a primary focus on zero-shot node classification tasks. Evaluations of few-shot node classification and link prediction tasks are provided in Appendix. D.6 D.7. First, we demonstrate the effectiveness of task-adaptive encoding and identify issues with existing methods that rely on aligning node embeddings with the LLM token space. Second, we validate the effectiveness of the proposed BP algorithm. Finally, we present the end-to-end performance of LLM-BP, comparing it to state-of-the-art baselines. We first introduce the datasets and baselines used in the study:

**Datasets** As listed in Table 1, we selected eleven real-world TAG datasets that encompass a variety of text domain shifts, including citation networks, e-commerce data, knowledge graphs, and webpage networks, which cover both homophily and heterophily structures. For more details, see Appendix B.1.

**Baselines:** We select representative baselines from all existing categories for model generalization on TAGs:

• *Vanilla LM / LLM Encoders*: including Sentence-BERT (SBert) (Reimers, 2019), RoBERTa (Liu, 2019), text-embedding-3-large (OpenAI, 2024), and bge-en-icl (Li et al., 2024a), a state-of-the-art LLM2Vec encoder.

• *Vanilla LLMs*: including GPT-3.5-turbo (Achiam et al., 2023) and GPT-4o (Hurst et al., 2024), the latter being among the most advanced LLMs in reasoning. They process raw node texts without incorporating graph structures.

• *Tuning LM Encoder / GNNs*: including ZeroG (Li et al.,

2024e), UniGLM (Fang et al., 2024) that tune LM encoders, ZeroG is specifically proposed for zero-shot node classification. DGI (Veličković et al., 2018), GraphMAE (Hou et al., 2022) that perform Graph-SSL are also compared.

• *LLMs with Graph Adapters*: including LLaGA (Chen et al., 2024b), TEA-GLM (Wang et al., 2024), and GraphGPT (Tang et al., 2024a), which are the three representative works adopting LLMs with projectors to align compressed node representations with the token space.

• *Multi-Task Graph Foundation Models*: Consisting of OFA (Liu et al., 2023a) and GOFA (Kong et al., 2024), which are the state-of-the-art multi-task foundation models.

• *LLMs for Data Augmentation*: referring to LLM-GNN (Chen et al., 2024e), specifically designed for zero-shot node classification, which utilizes LLMs as annotators for pseudo-labels and further train GNNs for inference.

• *Neighborhood Aggregation (NA)*: referring to the training-free method proposed in (Yang et al., 2024), which injects graph structural information into node representations by directly aggregating the averaged neighborhood embeddings.

**Settings:** Unlike LLM-BP which does not require additional fine-tuning of LLMs, most of the baselines–except from vanilla encoders, LLMs or NA–require fine-tuning. Methods of vanilla encoders and LLM-BP that require sampling nodes to obtain class embeddings under zero-shot settings are repeated 30 times with seed 42 to 71, and the average performance is reported in the following experiment sections. Implementation details for baselines and LLM-BP can be found in Appendix. B.3 B.2.

### 4.1. Evaluation for Task-Adaptive Node Embedding

#### • Exp.1: Ineffectiveness of LLMs w/ Graph Adapters

Figure 2 illustrates the accuracy of encoder-based methods alongside two representative LLMs-with-graph-adapters methods across each dataset. Notably, using text embeddings generated by SBert (Reimers, 2019) without incorporating graph structural information significantly outperforms both LLaGA (Chen et al., 2024b) and GraphGPT (Tang et al., 2024a). These two methods align node representations that combine SBert embeddings with graph information to the LLMs' token space via a projector. This finding suggests that the generalization capabilities of these approaches primarily stem from the pre-trained language model encoders rather than the LLMs' inherent understanding of TAG data. Consequently, future works should exercise caution when adopting this strategy.

#### • Exp.2: Effectiveness of The Task-Adaptive Encoder

According to Figure 2, the task-adaptive encoder achieves the best performance on most of the datasets, enhancing

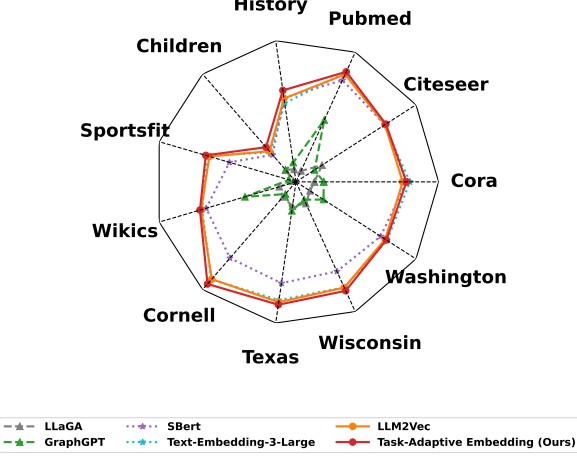

Figure 2: Zero-Shot Accuracy of vanilla encoders vs. LLMs-with-Graph-Adapters. All the encoder-based methods do not leverage graph structure information.

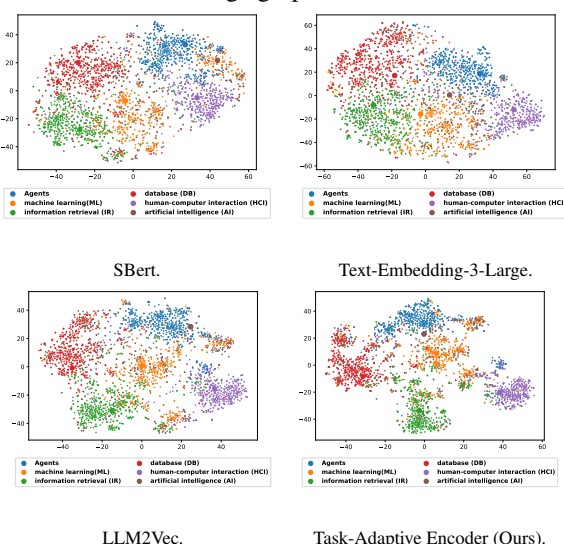

Figure 3: t-SNE visualization of encoders on Citeseer.

the vanilla LLM2Vec on average by $2.3\%$, highlighting the importance of incorporating task-specific information during encoding. To further illustrate this, we use the Citeseer (Giles et al., 1998) dataset as an example and perform t-SNE visualization (Van der Maaten & Hinton, 2008) on the embeddings derived from the encoders. As shown in Fig. 3, when provided with class information, the task-adaptive encoder generates embeddings that exhibit tighter clustering for the same class compared to other baselines. The significance test of improvement from task-adatove encoding is provided in Table. 5 in Appendix. D.1.

Note that the benefits of class information are observed only in encoders derived from LLM decoders potentially due to their strong contextual learning capabilities. As illustrated

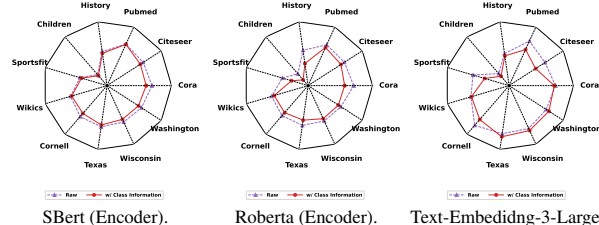

Figure 4: Class information fed into different encoders.

in Fig. 4, incorporating class information into smaller LM encoders, such as SBert (Reimers, 2019) or RoBERTa (Liu, 2019), may even degrade performance. Regarding Text-embedding-3-large (OpenAI, 2024), the impact of class information remains inconclusive due to the unknown internal mechanisms of the black-box encoder.

### 4.2. Generalizable Graph Aggregation

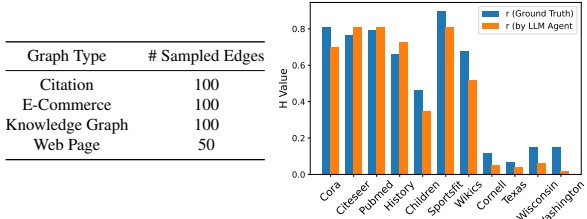

| Graph Type | # Sampled Edges |
|---|---|
| Citation | 100 |
| E-Commerce | 100 |
| Knowledge Graph | 100 |
| Web Page | 50 |

Figure 5: Left: Number of edges sampled per dataset. Right: GPT-4o-mini's prediction of the homophily level $r$.

● **Exp.3: LLM Agents for Homophily Level $r$ Estimation** As shown in Fig.5 (Left), we randomly sample $k$ edges ($k = 100$ for large graphs and $k = 50$ for small ones), incorporating them into prompts (Fig.1) for LLM-based estimation of the homophily level $r$ (Sec.3.4). We evaluate four LLMs: GPT-4o, GPT-4o-mini(Hurst et al., 2024), GPT-3.5-Turbo (Achiam et al., 2023), and Mistral-7B-Instruct v0.3 (Jiang et al., 2023). Each model responds to each node pair over five trials, with the final estimate determined by majority voting. Full results are provided in Fig.7 in appendix, demonstrating that GPT-4o-mini and GPT-4o effectively estimate $r$, GPT-3.5-Turbo performs reasonably well, while Mistral-7B-Instruct-v0.3 fails. Balancing accuracy and cost efficiency, we select GPT-4o-mini's estimation (Fig. 5 Right) for subsequent studies.

● **Exp.4: Effectiveness of the BP Algorithm** Experimental results are presented in Fig. 6, where we evaluate the four approaches over various graph structures. Specifically, We compare the BP algorithm (Eq. 6) and its linear approximation (Eq. 7) with vanilla encoders that do not utilize structure (Raw) and the NA baseline. For all the four encoders across all the datasets, the proposed BP algorithm slightly outperforms its linear approximation, and they consistently outperform Raw. Moreover, in most datasets, they

| | Homophilic | | | | | | | | | | | | | | Heterophilic | | | | | | | | Avg Rank | |
| | Citation Graph | | | | | | E-Commerce & Knowledge Graph | | | | | | | | Schools | | | | | | | | | |
| Method | Cora | | Citeseer | | Pubmed | | History | | Children | | Sportsfit | | Wikics | | Cornell | | Texas | | Wisconsin | | Washington | | | |
| | Acc | F1 | Acc | F1 | Acc | F1 | Acc | F1 | Acc | F1 | Acc | F1 | Acc | F1 | Acc | F1 | Acc | F1 | Acc | F1 | Acc | F1 | Acc | F1 |
|---|---|---|---|---|---|---|---|---|---|---|---|---|---|---|---|---|---|---|---|---|---|---|---|---|
| Sbert (Reimers, 2019) | 69.75 | 67.21 | 66.69 | 63.31 | 70.57 | 71.38 | 53.53 | 20.45 | 22.59 | 20.13 | 43.79 | 38.26 | 59.06 | 56.19 | 63.66 | 54.39 | 64.58 | 49.79 | 62.10 | 52.07 | 63.52 | 48.00 | 7.27 | 7.09 |
| Roberta (Liu, 2019) | 70.71 | 68.47 | 66.95 | 63.57 | 69.54 | 70.31 | 55.39 | 21.84 | 24.25 | 22.41 | 41.51 | 36.09 | 59.08 | 56.49 | 61.68 | 51.84 | 62.25 | 49.26 | 60.33 | 49.08 | 60.60 | 45.34 | 7.18 | 7.18 |
| Text-Embedding-3-Large (OpenAI, 2024) | 71.90 | 69.87 | 66.24 | 63.30 | 75.96 | 75.75 | 50.15 | 19.21 | 24.68 | 24.10 | 58.39 | 53.03 | 61.78 | 58.82 | 81.50 | 70.11 | 75.42 | 63.17 | 73.14 | 63.02 | 66.35 | **57.69** | 5.36 | 4.45 |
| LLM2Vec (BehnamGhader et al., 2024) | 67.34 | 65.92 | 67.13 | 64.37 | 74.57 | 74.65 | 53.14 | 19.06 | 25.56 | 24.31 | 57.00 | 52.29 | 62.34 | 58.32 | 81.26 | 69.08 | 76.68 | 63.12 | 73.36 | 62.50 | 65.92 | 53.34 | 5.64 | 5.36 |
| SBert + NA (Yang et al., 2024) | 72.49 | 69.90 | 68.66 | 64.75 | 71.26 | 71.87 | 57.86 | 21.98 | 25.28 | 22.74 | 46.84 | 40.85 | 66.26 | 63.57 | 54.21 | 44.66 | 56.04 | 41.09 | 54.23 | 46.11 | 58.88 | 43.05 | 5.82 | 6.00 |
| GPT-3.5-turbo (Achiam et al., 2023) | 70.11 | 52.11 | 66.83 | 47.58 | 89.75 | 66.16 | 55.07 | 30.36 | 29.73 | 26.13 | **67.21** | 54.45 | 65.53 | 51.19 | 45.54 | 39.30 | 56.14 | 32.53 | 58.86 | 46.84 | 51.09 | 35.68 | 5.64 | 8.18 |
| GPT-4o (Hurst et al., 2024) | 70.29 | 62.95 | 64.77 | 47.78 | **89.85** | 67.39 | 53.30 | **31.68** | **30.76** | **29.20** | 66.35 | 56.22 | 66.10 | 56.04 | 45.54 | 41.92 | 63.10 | 50.51 | 56.60 | 52.54 | 48.90 | 42.54 | 5.91 | 6.36 |
| UniGLM (Fang et al., 2024) | 45.57 | 43.25 | 52.26 | 48.41 | 70.33 | 69.78 | 44.24 | 24.84 | 21.48 | 19.17 | 33.46 | 32.99 | 55.05 | 52.08 | 23.03 | 22.06 | 21.39 | 18.90 | 27.16 | 26.45 | 24.01 | 23.08 | 11.36 | 9.91 |
| ZeroG (Li et al., 2024e) | 60.4 | 56.02 | 50.35 | 45.15 | 74.68 | 71.75 | 36.55 | 16.84 | 12.72 | 12.61 | 14.27 | 5.33 | 46.74 | 40.86 | 10.47 | 6.46 | 53.48 | 15.95 | 12.66 | 5.02 | 8.3 | 3.07 | 12.27 | 12.73 |
| DGI (Veličković et al., 2018) | 16.79 | 12.77 | 15.24 | 15.04 | 25.10 | 19.18 | 20.98 | 3.89 | 2.22 | 1.04 | 7.48 | 3.47 | 14.98 | 4.24 | 14.66 | 10.02 | 11.23 | 9.42 | 12.08 | 6.95 | 20.96 | 14.15 | 13.91 | 14.73 |
| GraphMAE (Hou et al., 2022) | 15.13 | 7.10 | 8.11 | 7.67 | 36.56 | 34.29 | 36.36 | 5.75 | 7.24 | 1.97 | 30.50 | 6.99 | 8.91 | 4.03 | 23.04 | 14.95 | 17.65 | 11.67 | 23.02 | 11.87 | 24.89 | 13.34 | 15.18 | 15.45 |
| OFA (Liu et al., 2023a) | 20.36 | 16.57 | 41.31 | 33.37 | 28.18 | 26.62 | 8.25 | 3.48 | 3.05 | 2.29 | 15.18 | 4.7 | 30.77 | 25.22 | 29.84 | 12.62 | 11.77 | 5.87 | 4.8 | 3.44 | 6.04 | 4.28 | 13.91 | 14.73 |
| GOFA (Kong et al., 2024) | 71.06 | 70.21 | 65.72 | 64.18 | 74.76 | 73.00 | 56.25 | 31.57 | 12.15 | 7.73 | 37.87 | 33.19 | **68.62** | 62.93 | 39.50 | 35.47 | 38.37 | 29.54 | 32.51 | 25.12 | 31.02 | 21.24 | 8.00 | 7.45 |
| GraphGPT (Tang et al., 2024a) | 17.48 | 12.68 | 13.93 | 12.78 | 42.94 | 25.68 | 12.31 | 9.15 | 9.94 | 4.24 | 4.53 | 2.44 | 33.59 | 30.21 | 10.18 | 14.71 | 18.48 | 9.85 | 12.35 | 6.32 | 20.64 | 15.79 | 14.55 | 14.64 |
| LLAGA (Chen et al., 2024b) | 11.62 | 14.42 | 19.52 | 23.34 | 7.56 | 13.42 | 7.95 | 8.89 | 10.09 | 5.02 | 1.84 | 2.66 | 10.98 | 16.73 | 12.57 | 20.1 | 15.51 | 22.97 | 15.09 | 20.85 | 10.48 | 18.98 | 15.36 | 13.64 |
| LLM-BP | **72.59** | **71.10** | **69.51** | **66.29** | 75.55 | 75.32 | **59.86** | 22.66 | 24.81 | 22.66 | 61.92 | **57.51** | 67.75 | 63.53 | 83.28 | 71.80 | **81.66** | **65.41** | **77.75** | **63.70** | **73.14** | 57.33 | **2.27** | 2.55 |
| LLM-BP (appr.) | 71.41 | 70.11 | 68.66 | 65.62 | 76.81 | **76.81** | 59.49 | 23.02 | 29.40 | 28.45 | 61.51 | 57.09 | 67.96 | **64.27** | **84.92** | **74.19** | 79.39 | 64.63 | 75.65 | 62.53 | 70.04 | 55.53 | 2.45 | **2.27** |

Table 2: Zero-Shot End-to-End Evaluation. 'NA' refers to neighborhood embedding aggregation.

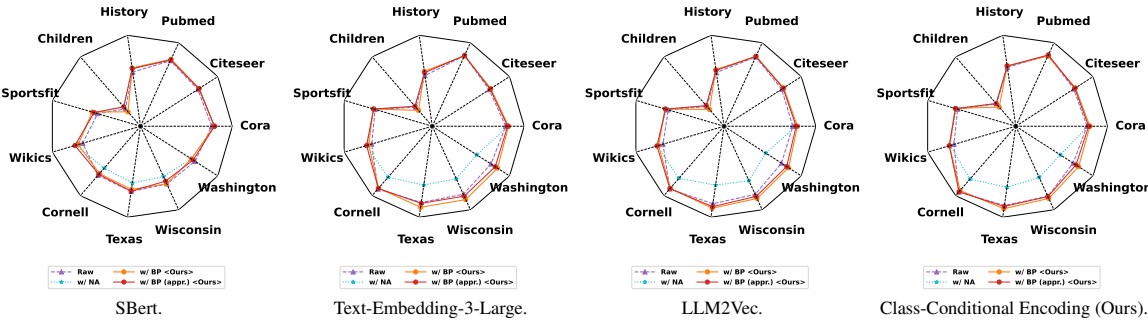

| SBert. | Text-Embedding-3-Large. | LLM2Vec. | Class-Conditional Encoding (Ours). |

Figure 6: Experiments on graph information aggregation. 'Raw' refers no graph structure usage, 'w/ NA' refers to the neighborhood embedding aggregation (NA) proposed in (Yang et al., 2024), 'w/ BP' refers to the belief propagation following Eq. 6, 'w/ BP (appr.)' refers to its simplified linear form that follows Eq. 7.

also surpass the NA baseline, particularly on heteophilic graphs, where direct neighborhood embedding aggregation negatively affects performance. These results highlight the generalizability of our data modeling approach and the effectiveness of the key-parameter estimation design in BP.

### 4.3. End-to-End Evaluation

• **Exp.5: Main Results in the Zero-Shot Setting** The main experimental results are presented in Table 2. Among the baselines, vanilla encoders and LLMs demonstrate strong zero-shot generalization. GPT-3.5-Turbo (Achiam et al., 2023) ranks first on the Sportsfit dataset, while GPT-4o (Hurst et al., 2024) achieves the best performance on Pubmed and Children.

UniGLM (Fang et al., 2024) and ZeroG (Li et al., 2024e) perform well in domains aligned with their pre-training, such as citation networks (e.g., ZeroG enhances SBert's performance on Cora, Pubmed, and Wikics). However, both struggle on TAGs with unseen text distributions (e.g., Sportsfit) or novel graph structures (e.g., webpage networks), suggest-

ing that fine-tuned LM encoders may suffer performance degradation on out-of-domain TAGs. Similarly, graph-SSL methods (DGI (Veličković et al., 2018), GraphMAE (Hou et al., 2022)) show limited generalization across structural shifts.

Among multi-task graph foundation models, GOFA achieves strong performance, likely benefiting from a larger pre-training corpus for graph-text alignment (Hu et al., 2021; Ding et al., 2023) compared to GraphGPT (Tang et al., 2024a) and LLaGA (Chen et al., 2024b), which are trained solely on ogbn-arxiv. However, GOFA still requires broader pre-training and instruction fine-tuning to improve generalization under text domain shifts, and its reliance on GNNs may limit effectiveness on heterophilic data.

Notably, LLM-BP and LLM-BP (appr.) achieve the highest average ranking across all datasets on both homophilic and heterophilic graphs. For fine-grained average ranking that distinguish between homophilic and heterophilic graphs, refer to Appendix D.2. Another interesting observation is that when we randomly sample $20c$ nodes to obtain the class

embeddings with the help of LLMs following Algorithm. 1, the zero-shot performance of the encoders in this setting is comparable to their performance between 5-10-shot setting as shown in Table. 10 in the Appendix. Further comparisons with LLM-GNN (Chen et al., 2024e) and TEA-GLM (Wang et al., 2024) are provided in Appendix D.5.

• **Exp.6: Main Results under Few-shot Setting** We conduct the evaluation in $k = 1, 3, 5, 10$-shot settings. Using 10 different random seeds, we sample the shots from the training set and repeat the experiments 10 times. The experimental results are presented in Table 10 in Appendix D. Across all $k$-shot settings, LLM-BP and LLM-BP (appr.) outperform the baseline models.

## 5. Discussion and Limitations

Graph learning tasks often face substantial data constraints compared to other domains, underscoring the importance of establishing fundamental principles that foster model generalization. Our approach exemplifies this by leveraging LLMs to analyze graph data and determine suitable inference strategies, particularly via homophily estimation for belief propagation. While LLM-BP achieves notable success on TAGs for node classification and extends partially to link prediction, it remains a step away from a fully comprehensive graph foundation model that addresses a wider range of graph learning tasks. Nonetheless, the core idea of leveraging LLM-driven graph analysis to guide algorithmic decisions aligned with task-specific inductive biases holds broad potential for future applications.

## Acknowledgements

H. Wang, S. Liu, R. Wei and P. Li are partially supported by NSF awards IIS-2239565, CCF-2402816, IIS-2428777, PHY-2117997; DOE award DE-FOA-0002785; JPMC faculty awards; Openai Research Credits; and Meta research award.

We extend our sincere gratitude to Hongbin Pei, Zhen Wang, and Jingcheng Cen for their valuable assistance in identifying the raw node text of the heterophilic graphs used in this study.

## Impact Statement

This paper presents work whose goal is to advance the field of Machine Learning. There are many potential societal consequences of our work, none which we feel must be specifically highlighted here.

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

# A. More Related Works

**LLMs for Data Augmentation** annotate pseudo-labels via their advanced zero-shot text classification performance. *E.g.*, LLM-GNN (Chen et al., 2024e), Cella (Zhang et al., 2024b) and (Hu et al., 2024) propose heuristics to actively select and annotate pseudo-labels for supervised GNN training. (Pan et al., 2024) performs knowledge distillation with LLMs as teachers. (Yu et al., 2023; Li et al., 2024c) generate synthetic node text with LLMs. The performance of these methods depend on the capability of LLM, and may still require relatively high annotating and training cost.

**LLMs for Graph Property Reasoning** focus on reason graph structure properties (e.g., shortest path, node degree, etc) (Tang et al., 2024b; Dai et al., 2024; Yuan et al., 2024; Ouyang et al., 2024). Representative works include (Perozzi et al., 2024; Chen et al., 2024a; Zhang et al., 2024c; Cao et al., 2024; Wei et al., 2024).

**Tuning LMs/GNNs towards Better Task-Specific Performance** aims to push the limits of task-specific performance on TAGs other than generalization. These methods develop novel techniques to optimize LMs or GNNs for pushing the limits of in-domain performance (Chien et al., 2022; Duan et al., 2023; He et al., 2024; Zhao et al., 2023b; Zhu et al., 2021; Li et al., 2021; Yang et al., 2021; Bi et al., 2021; Pang et al., 2022; Zolnai-Lucas et al., 2024; Yang et al., 2021).

**Text embeddings** Generating unified text embeddings is a critical research area with broad applications, including web search, accounting documents (Li et al., 2024b) and question answering. Numerous text encoders (Reimers, 2019; Liu, 2019; Song et al., 2020) based on pre-trained language models have served as the foundation for various embedding models. Recently, decoder-only LLMs have been widely adopted for text embedding tasks (Li et al., 2023; Moreira et al., 2024) achieving remarkable performance on the Massive Text Embedding Benchmark (MTEB) (Muennighoff et al., 2022). This progress stems from LLM2Vec (BehnamGhader et al., 2024), which introduces a novel unsupervised approach to transforming decoder-only LLMs into embedding models, including modifications to enable bidirectional attention. Recent findings (Li et al., 2024a) suggest that retaining the unidirectional attention mechanism enhances LLM2Vec's empirical performance.

# B. Experiment Details

### B.1. Dataset Details

**Meta-Data** In Table. 3, we show the meta-data of all the eleven datasets used in our experiments.

|  | Number of Nodes | Number of Edges | Number of Classes | Ground Truth Homophily Ratio |
|---|---|---|---|---|
| Cora | 2708 | 10556 | 7 | 0.809 |
| Citeseer | 3186 | 8450 | 6 | 0.764 |
| Pubmed | 19717 | 88648 | 3 | 0.792 |
| History | 41551 | 503180 | 12 | 0.662 |
| Children | 76875 | 2325044 | 24 | 0.464 |
| Sportsfit | 173055 | 3020134 | 13 | 0.9 |
| Wikics | 11701 | 431726 | 10 | 0.678 |
| Cornell | 191 | 292 | 5 | 0.115 |
| Texas | 187 | 310 | 5 | 0.067 |
| Wisconsin | 265 | 510 | 5 | 0.152 |
| Washington | 229 | 394 | 5 | 0.149 |

Table 3: Meta data of the datasets in this study.

**Dataset Split** For the datasets (all the homophily graphs) that have been used for study in TSGFM (Chen et al., 2024d), we follow their implementation to perform data pre-processing, obtain raw texts and do data split, the introduction to data source can be found at Appendix.D.2 in their original paper, the code can be found at the link [1].

As to the heterophily graphs, the four datasets are originally from (Craven et al., 1998). We obtain the raw texts from (Yan

---

[1] https://github.com/CurryTang/TSGFM/tree/master?tab=readme-ov-file

et al., 2023), which can be found from[2]. As to data split, for zero-shot inference, all the nodes are marked as test data; for few-shot setting, $k$ labeled nodes are randomly sampled per class and the rests are marked as test data. To the best of our knowledge, the four heterophily graph datasets used in this study are the only graphs that provide raw texts feature.

## B.2. LLM-BP Implementation Details

**Infrastructure and Seeds** All the local experiments run on a server with AMD EPYC 7763 64-Core Processor and eight NVIDIA RTX 6000 Ada GPU cards, methods are mainly implemented with PyTorch (Paszke et al., 2019), Torch-Geometric (Fey & Lenssen, 2019) and Huggingface Transformers (Wolf, 2019). To obtain the embeddings, all the encoders that run locally on the server without API calling in this study run with the random seed 42.

**Class Embedding**

• **Zero-Shot Setting:** We uniformly randomly sample $20c$ nodes per graph, where $c$ denotes the number of classes, we employ GPT-4o (Hurst et al., 2024) to infer their labels. With the predictions from LLMs, the sampled nodes form distinct clusters. For each cluster, we take the top-$k$ (10 in the experiments) nodes whose embedding are closest with the cluster center and calculate their average embedding as the class embedding.

We notice that some works directly feed text descriptions into encoders as class embeddings (Yang et al., 2024; Chen et al., 2024d), we find that different encoders can be highly sensitive to variations in text description. Therefore, we adopt the above method to ensure fairness among different encoders.

•**Few-Shot Setting:** We directly take the class embedding as the averaged embeddings of labeled nodes per class.

**The Task-Adaptive Encoder:** We directly adopt the pre-trained LLM2Vec encoder released by (Li et al., 2024a), which is based on Mistral7B-v0.1 (Jiang et al., 2023). We check the pre-training data used in the original paper for aligning LLM decoders with the embedding space, the datasets are mainly for text-retrieval and therefore do not overlap with the TAG datasets adopted in our study. For detailed introduction of the datasets for LLM2Vec pre-training, see Section 4.1 training data in the original paper. The task-adaptive prompting follows the format as:

⟨ *Instruct* ⟩
*"Given the {task description}, classify it into one of the following $k$ classes:*
*{class labels}*
⟨ *query*⟩
*{raw node texts}."*
⟨ *response* ⟩

, where the {task descriptions} prompts for each dataset is the same as that used for vanilla LLMs, see Table. 13 for details.

**Hyper-Parameters for BP algorithm** For LLM-BP, we adopt 5 message-passing layers, for its linear approximation form, we use a single layer. The temperature hyper-parameter $\tau$ in computing node potential initialization in Eq. (5) is set as 0.025 for LLM-BP and 0.01 for LLM-BP (appr.) across all the datasets. Attached in Table. 4 is the homophily ratio $r$ we used (predicted by GPT-4o-mini (Hurst et al., 2024).

| | Cora | Citeseer | Pubmed | History | Children | Sportsfit | Wikics | Cornell | Texas | Wisconsin | Washington |
|---|---|---|---|---|---|---|---|---|---|---|---|
| Ground Truth Homophony Ratio | 0.81 | 0.76 | 0.79 | 0.66 | 0.46 | 0.90 | 0.67 | 0.11 | 0.06 | 0.15 | 0.19 |
| $r$ predicted by GPT-4o-mini | 0.70 | 0.81 | 0.81 | 0.73 | 0.35 | 0.81 | 0.52 | 0.05 | 0.04 | 0.06 | 0.02 |

Table 4: $r$ predicted by GPT-4o-mini, that is used in all the experiments in this study.

---

[2]https://github.com/sktsherlock/TAG-Benchmark/tree/master

## B.3. Baseline Implementation Details

• **Vanilla Encoders** Vanilla encoders like SBert (Reimers, 2019), Roberta (Liu, 2019) and text-embedding-3-large (OpenAI, 2024) directly encode the raw text of the nodes. LLM2Vec uses the prompts:

$$\langle\text{Instruct}\rangle\{\text{task\_description}\}\langle\text{query}\rangle X_i \langle\text{response}\rangle. \tag{8}$$

, where the {task\_description} for each dataset is provided in Appendix. E.1.

• **Vanilla LLMs** Prompts for GPT-4o and GPT-3.5-turbo adopts the format as follows:

*"role": "system"*
*"content": "You are a chatbot who is an expert in text classification"*
*"role": "user"*
*"content": "We have {task description} from the following k categories: {class labels}*
*The text is as follows:*
*{raw node text}*
*Please tell which category the text belongs to:"*

The {task description} for the vanilla LLMs for each class is provided in Appendix. E.2.

• **Tuning LM/GNNs** We adopt the pre-trained UniGLM (Fang et al., 2024) released by the official implementation, which adopts Bert as the encoder, for direct inference. For ZeroG (Li et al., 2024e), we re-implement the method and train it on ogbn-arxiv (Hu et al., 2020) for fair comparison with other baselines.

As to GNNs tuning methods, we pre-train GraphMAE (Hou et al., 2022) and DGI (Veličković et al., 2018) on ogbn-arxiv (Hu et al., 2020), where the input for both models are from SBert (Reimers, 2019), and we follow in implementation in TSGFM (Chen et al., 2024d) benchmark.

• **Multi-Task GFMs** OFA (Liu et al., 2023a) is trained on ogbn-arxiv (Hu et al., 2020). As to GOFA, we directly adopt the model after pre-training (Hu et al., 2021; Ding et al., 2023) and instruct fine-tuning on ogbn-arxiv (Hu et al., 2020) provided by the authors due to the huge pre-training cost, the zero-shot inference scheme also follows their original implementation.

• **LLMs with Graph Adapters** Both LLaGA (Chen et al., 2024b) and GraphGPT (Tang et al., 2024a) are trained on ogbn-arxiv (Hu et al., 2020), we follow the hyper-parameter setting in their original implementation.

# C. Detailed Derivations

## C.1. Derivation for Eq. (4)

In a node classification task, given a node $i$, our goal is to minimize the mean-square error (MSE) in predicting the node label under the observations $\boldsymbol{X}$:

$$\min \text{MSE}(\hat{y}_i) = \mathbb{E}\left[(y_i - \hat{y}_i)^2 \mid \boldsymbol{X}\right]. \tag{9}$$

The optimal solution $\hat{y}_i$ is then given by:

$$\hat{y}_i = \sum_{y_i} y_i\, p(y_i \mid \boldsymbol{X}), \tag{10}$$

where the posterior marginal $p(y_i \mid \boldsymbol{X})$ is computed as:

$$p(y_i \mid \boldsymbol{X}) = \sum_{Y \setminus i} \mathbb{P}(Y \mid \boldsymbol{X}). \tag{11}$$

**Factorized Posterior under an MRF.** Assuming a Markov Random Field (MRF) over a graph $\mathcal{G} = (\mathcal{V}, \mathcal{E})$, the posterior distribution factors as:

$$\mathbb{P}_{\mathcal{G}}(Y \mid \boldsymbol{X}) \propto \prod_{i \in \mathcal{V}} \varphi_{X_i}(y_i) \prod_{(i,j) \in \mathcal{E}} \psi_{ij}(y_i, y_j), \tag{12}$$

where the node potential is defined as $\varphi_{X_i}(y_i) = \varphi_{y_i}(X_i)\phi_i(y_i)$.

**General Message-Passing Framework.** To compute the marginal $p(y_i \mid \boldsymbol{X})$, the loopy belief propagation (LBP) algorithm iteratively updates messages between nodes. The general message update rule from node $i$ to node $j$ at iteration $k$ is:

$$m_{i \rightarrow j}^{(k)}(y_j) = \alpha_{i \rightarrow j} \sum_{y_i} \left[ \varphi_{X_i}(y_i) \psi_{ij}(y_i, y_j) \prod_{\ell \in \mathcal{N}(i) \backslash j} m_{\ell \rightarrow i}^{(k-1)}(y_i) \right], \tag{13}$$

where $\alpha_{i \rightarrow j}$ is a normalization constant ensuring the message sums to 1.

**Node Belief Updates.** The node belief $p_i^{(k)}(y_i)$ at iteration $k$ is obtained by combining the node potential with incoming messages from all neighbors:

$$p_i^{(k)}(y_i) = \varphi_{X_i}(y_i) \prod_{\ell \in \mathcal{N}(i)} m_{\ell \rightarrow i}^{(k)}(y_i). \tag{14}$$

**Reformulating the Messages.** Substituting Eq. (14) into Eq. (13) simplifies the message-passing equation. The message from node $i$ to node $j$ at iteration $k$ can be rewritten as:

$$m_{i \rightarrow j}^{(k)}(y_j) = \alpha_{i \rightarrow j} \sum_{y_i} \psi_{ij}(y_i, y_j) \frac{p_i^{(k)}(y_i)}{m_{j \rightarrow i}^{(k-1)}(y_i)}. \tag{15}$$

This reformulation prevents double-counting the contribution of node $j$ to node $i$ in the previous iteration.

**Log-Space Stability.** To avoid numerical underflow, the log-space version of the message update is commonly used:

$$\log m_{i \rightarrow j}^{(k)}(y_j) = \text{LSE}_{y_i} \left[ \log \psi_{ij}(y_i, y_j) + \log p_i^{(k)}(y_i) - \log m_{j \rightarrow i}^{(k-1)}(y_i) \right], \tag{16}$$

where $\text{LSE}(\cdot) \equiv \log \sum \exp(\cdot)$.

**Summary.** By iteratively applying these message updates and node belief calculations, LBP provides an approximation for the posterior marginal $p(y_i \mid \boldsymbol{X})$. The final prediction $\hat{y}_i$ under the MMSE criterion is:

$$\hat{y}_i = \sum_{y_i} y_i \, p_i^{(k)}(y_i). \tag{17}$$

This completes the derivation of the message-passing update in Eq. (4).

# D. More Experiment Results

## D.1. Significance Test of Effectiveness of Task-Adaptive Encoding

| | | Cora | Citeseer | Pubmed | History | Children | Sportsfit | Wikics | Cornell | Texas | Wisconsin | Washington |
|---|---|---|---|---|---|---|---|---|---|---|---|---|
| Task-Adaptive Encoding vs. Vanilla LLM2Vec | 90% CI low, high | 0.7% 1.3% | -0.2% 1.3% | 1.3% 2.3% | 3.2% 5.2% | 3.3% 3.7% | 1.0% 2.6% | 1.0% 1.8% | 4.0% 4.9% | 1.5% 2.7% | 0.1% 0.8% | -0.2% 0.5% |
| | P value | **1e-8** | 0.38 | **1e-10** | **1e-9** | **3e-59** | **1e-4** | **3e-8** | **7e-18** | **5e-8** | **1e-3** | 0.82 |
| Task-Adaptive Encoding vs. Text-Embedding-3-Large | 90% CI low, high | -0.3% -0.2% | 0.5% 1.0% | -0.3% 1.0% | 6.9% 9.1% | 4.1% 4.4% | 0.7% 2.8% | 1.1% 2.0% | 3.1% 4.1% | 3.1% 4.7% | 0.1% 1.2% | -0.04% -0.1% |
| | P value | 6e-27 | **8e-7** | 0.51 | **7e-21** | **5e-59** | **0.03** | **1e-9** | **2e-25** | **1e-9** | 0.07 | 1e-13 |

Table 5: Lower and upper bound of improvement ($\uparrow$) in accuracy of task-adaptive encoding over baselines with 90% confidence interval, with significance level $p$ ($\downarrow$).

We conduct significance test on the improvment of task-adaptive encoding over vanilla LLm2Vec (Li et al., 2024a) and Text-Embedding-3-Large (OpenAI, 2024) under the zero-shot setting, with results shown in Table. 5. We replicate experiment for 100 times with random seeds from 42 to 141 and obtain classification accuracy of each method. To check normality, we

first apply Shapiro-Wilk test (SHAPIRO & WILK, 1965). If the data follows a normal distribution, we perform a Paired-t test (Student, 1908); otherwise, we use Wilcoxon Signed-Rank test (Wilcoxon, 1992), with packages from SciPy (Virtanen et al., 2020). The lower and upper bounds under 90% confidence interval are estimated with bootstrap algorithm (Tibshirani & Efron, 1993) to sample 10,000 times. Task-adaptive encoding show statistically significant improvement over vanilla LLM2Vec in 9 out of 11 dataset and outperforms Text-Embedding-3-Large in 8 out of 11 datasets (bolded in the table).

### D.2. Fine-Grained Ranking Results

We report the fine-grained ranking results of each method on homophilic and heterophilic graphs. The ranking is shown in Table. 6. Across the three sub-categories of graphs, LLM-BP and its approximation algorithm both achieves the top performance.

| Ranking | Homophilic | | Heterophilic | |
|---|---|---|---|---|
| | Acc | F1 | Acc | F1 |
| Sbert | 8.57 | 8.00 | 5.00 | 5.50 |
| Roberta | 7.71 | 7.57 | 6.25 | 6.50 |
| Text-Embedding-3-Large | 6.43 | 5.71 | 3.50 | 2.25 |
| LLM2Vec | 6.86 | 6.14 | 3.50 | 4.00 |
| SBert + NA | 4.57 | 5.00 | 8.00 | 7.75 |
| GPT-3.5-turbo | 4.43 | 7.86 | 7.75 | 8.75 |
| GPT-4o | 4.86 | 6.29 | 7.75 | 6.50 |
| UniGLM | 11.00 | 9.43 | 12.00 | 10.75 |
| ZeroG | 11.29 | 11.00 | 14.00 | 15.75 |
| DGI | 15.29 | 15.71 | 15.00 | 15.00 |
| GraphMAE | 14.86 | 15.29 | 12.25 | 13.75 |
| OFA | 14.29 | 14.29 | 15.25 | 16.25 |
| GOFA | 6.71 | 5.71 | 10.25 | 10.50 |
| GraphGPT | 14.43 | 14.86 | 14.75 | 14.25 |
| LLAGA | 15.86 | 14.71 | 14.50 | 11.75 |
| LLM-BP | **2.86** | 3.14 | **1.25** | **1.50** |
| LLM-BP (appr.) | **2.86** | **2.29** | 1.75 | 2.25 |

Table 6: Average ranking in homophilic and homophilic graphs.

### D.3. LLM Agents' Prediction on Homophily Ratio $r$

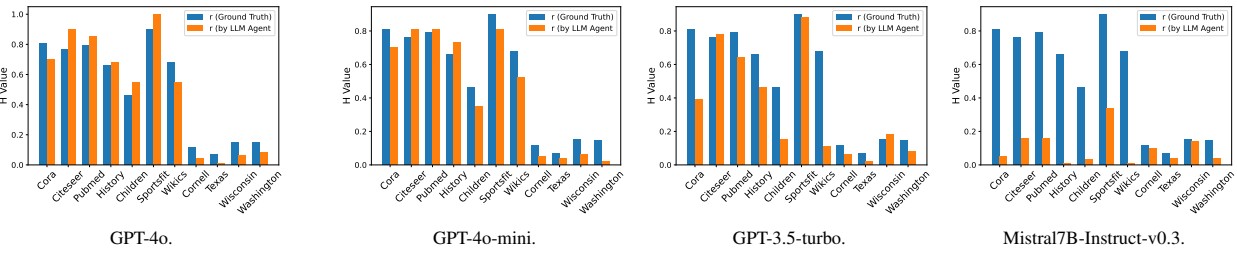

Figure 7: LLM agents' performance on predicting the homophily constant $r$.

More prediction performance of GPT-4o, GPT-3.5-turbo and Mistral7b-Instruct-v3 are shown in Fig. 7.

### D.4. Sensitivity Analysis of Sampled Edge Numbers

We conduct sensitivity analysis on homophily ratio $r$ prediction with respect to the number of sampled edges that feeds to LLM. According to Table. 7, the homophily ratio prediction performance is stable across the sampled edge numbers from 40 to 100.

| | Cora | | Citeseer | | Pubmed | | Bookhis | | Bookchild | | Sportsfit | | Wikics | |
|---|---|---|---|---|---|---|---|---|---|---|---|---|---|---|
| | value | gap ↓ | value | gap ↓ | value | gap ↓ | value | gap ↓ | value | gap ↓ | value | gap ↓ | value | gap ↓ |
| ground truth | 0.81 | - | 0.76 | - | 0.79 | - | 0.66 | - | 0.46 | - | 0.90 | - | 0.67 | - |
| 100 | 0.70 | 0.11 | 0.81 | 0.05 | 0.81 | 0.02 | 0.73 | 0.07 | 0.35 | 0.11 | 0.81 | 0.09 | 0.52 | 0.15 |
| 80 | 0.70 | 0.11 | 0.77 | 0.01 | 0.83 | 0.04 | 0.75 | 0.09 | 0.37 | 0.09 | 0.76 | 0.14 | 0.55 | 0.12 |
| 40 | 0.65 | 0.16 | 0.77 | 0.01 | 0.81 | 0.02 | 0.75 | 0.09 | 0.33 | 0.13 | 0.75 | 0.15 | 0.50 | 0.17 |

Table 7: Sensitivity test of homophily ratio prediction performance with respect to the number of sampled edges. As shown in the left column, 100, 80 and 40 edges are sampled to feed the LLMs to predict homophily ratio $r$. 'value' refers to the predicted $r$, and 'gap' is the gap between prediction and ground truth.

## D.5. Zero-Shot Comparison with LLM-GNN (Chen et al., 2024e) and TEA-GLM (Wang et al., 2024)

| | Cora | Citeseer | Pubmed | Wikics |
|---|---|---|---|---|
| DA-AGE-W | 74.96 | 58.41 | 65.85 | 59.13 |
| DA-RIM-W | 74.73 | 60.80 | 77.94 | 68.22 |
| DA-GraphPart-W | 68.61 | 68.82 | 79.89 | 67.13 |
| LLM-BP | 72.59 | 69.51 | 75.55 | 67.75 |
| LLM-BP (app.) | 71.41 | 68.66 | 76.81 | 67.96 |

Table 8: Accuracy compared with LLM-GNN, where 'DA' denotes the 'C-Density' methods proposed in (Chen et al., 2024e), '-W' refers to the weighted cross-entropy loss function used for training, AGE (Cai et al., 2017), RIM (Zhang et al., 2021), GraphPart (Ma et al., 2022) are different graph active learning baselines used in the original paper.

| | Cora | Pubmed | History | Children |
|---|---|---|---|---|
| TEA-GLM | 20.2 | 84.8 | 52.8 | 27.1 |
| LLM-BP | 72.59 | 75.55 | 59.86 | 24.81 |
| LLM-BP (app.) | 71.41 | 76.81 | 59.49 | 29.4 |

Table 9: Accuracy compared with TEA-GLM (Wang et al., 2024).

Here we present the comparison with LLM-GNN (Chen et al., 2024e) in Table. 8. We compare with three different graph active learning heuristics from their original paper. Our training-free methods, LLM-BP and LLM-BP (appr.) achieves top performance on Citeseer and Wikics, while performs comparably with the baselines in Cora and Pubmed. Note that the results of LLM-GNN are from Table. 2 in the original paper.

The comparison with TEA-GLM is shown in Table. 9. Results of TEA-GLM are from Table.1 in their original paper.

## D.6. Experiment Results in Few-Shot Setting

We use 10 different random seeds from 42 to 52 to sample the $k$-shot labeled nodes from training dataset, and report the average accuracy and macro $F1$ score with standard variance. Results are shown in Table. 10. Across all the $k$s, our LLM-BP achieves the top ranking performance across all the eleven datasets, exhibiting similar insights with the zero-shot setting.

## D.7. Zero-Shot Link Prediction Results

For each dataset, We randomly sample & remove 1000 edges and 1000 node pairs from the graph as testing data. A straightforward approach is to compare the cosine similarity between node embeddings to determine the presence of a link. Specifically, we aggregate embeddings for 3 layers on the incomplete graph and compute the cosine similarity between node representations, achieving better zero-shot performance than LLMs-with-Graph-Adapters methods (Wang et al., 2024; Chen et al., 2024b; Tang et al., 2024a), as shown in Table. 11. Note that the performance in the table refers to LLM-with-Graph-Adapters that have only been trained on other tasks and never on link prediction tasks.

| | | Cora | | Citeseer | | Pubmed | | History | | Children | | Sportsfit | | Wikics | | Cornell | | Texas | | Wisconsin | | Washington | | Avg. Rank | |
|---|---|---|---|---|---|---|---|---|---|---|---|---|---|---|---|---|---|---|---|---|---|---|---|---|---|
| | | Acc | F1 | Acc | F1 | Acc | F1 | Acc | F1 | Acc | F1 | Acc | F1 | Acc | F1 | Acc | F1 | Acc | F1 | Acc | F1 | Acc | F1 | Acc | F1 |
| **1-Shot** | | | | | | | | | | | | | | | | | | | | | | | | | |
| SBert | Raw | 42.8±5.2 | 42.1±5.7 | 42.4±7.7 | 38.8±6.6 | 52.2±6.7 | 51.1±6.6 | 30.6±7.3 | 14.9±2.5 | 10.5±3.2 | 9.1±1.7 | 17.8±4.4 | 17.2±3.0 | 34.6±3.2 | 31.0±3.1 | 40.4±9.1 | 32.3±6.9 | 26.3±9.4 | 23.5±6.0 | 47.0±7.7 | 35.1±5.0 | 36.0±12.8 | 27.6±8.4 | 9.9 | 10.7 |
| SBert | BP | 43.9±5.1 | 43.0±5.6 | 43.8±8.1 | 39.9±7.0 | 53.6±6.8 | 52.2±6.7 | 33.0±8.2 | 15.5±2.6 | 9.8±2.7 | 8.4±1.4 | 18.7±5.1 | 18.3±3.5 | 35.9±3.5 | 32.0±3.5 | 41.3±8.9 | 33.4±6.5 | 27.9±8.1 | 25.6±5.7 | 47.8±8.4 | 35.8±5.2 | 35.9±11.4 | 27.9±7.3 | 8.5 | 8.6 |
| SBert | BP (appr.) | 43.1±4.9 | 42.4±5.4 | 42.8±7.7 | 39.1±6.6 | 52.5±6.7 | 51.3±6.6 | 31.2±7.6 | 15.1±2.6 | 10.6±3.3 | 9.2±1.6 | 18.0±4.5 | 17.4±3.1 | 35.2±3.3 | 31.4±3.2 | 40.9±9.0 | 32.9±6.9 | 26.7±9.1 | 24.1±5.9 | 47.3±7.9 | 35.2±5.0 | 36.0±12.3 | 27.8±8.1 | 8.9 | 9.5 |
| T-E-3-Large | Raw | 47.2±5.4 | 46.3±5.7 | 40.2±6.3 | 37.0±5.5 | 55.2±7.3 | 52.8±7.2 | 22.9±7.1 | 12.5±1.9 | 13.3±2.7 | 12.0±1.4 | 33.1±6.6 | 30.6±4.3 | 40.3±4.6 | 36.4±3.4 | 55.4±8.4 | 46.5±7.0 | 50.8±7.7 | 42.5±6.4 | 58.6±5.2 | 47.8±4.3 | 44.2±16.9 | 36.1±10.6 | 7.2 | 7.1 |
| T-E-3-Large | BP | **49.1±5.3** | **48.0±5.5** | 41.6±7.0 | 38.0±6.1 | 55.8±8.6 | 52.7±8.5 | 24.1±8.0 | 13.1±2.1 | 12.1±2.3 | 10.8±1.3 | 34.7±4.7 | 32.1±4.8 | **42.4±5.0** | **38.0±3.7** | 59.6±6.8 | 50.7±6.6 | 52.9±8.2 | 44.5±6.5 | 59.8±5.6 | 49.0±4.5 | 45.5±15.8 | 37.9±10.4 | 5.2 | 5.1 |
| T-E-3-Large | BP (appr.) | 48.4±5.2 | 47.4±5.6 | 40.7±6.6 | 37.4±5.7 | 55.4±7.7 | 52.8±7.5 | 23.3±7.4 | 12.7±2.0 | 13.4±2.7 | 12.2±1.4 | 33.5±6.8 | 30.9±4.4 | 41.3±4.8 | 37.3±3.5 | 57.0±7.3 | 47.7±6.8 | 51.8±8.0 | 43.4±6.6 | 59.0±5.0 | 48.5±4.5 | 44.3±16.6 | 36.3±10.2 | 6.1 | 6.0 |
| LLM2Vec | Raw | 39.8±6.6 | 38.0±6.9 | 47.9±9.5 | 42.2±7.7 | 54.9±7.7 | 52.5±7.9 | 24.2±12.9 | 12.8±3.6 | 10.0±1.4 | 9.8±1.3 | 27.6±5.6 | 26.1±4.9 | 33.9±4.1 | 31.3±3.8 | 53.0±9.9 | 43.8±6.6 | 52.9±11.2 | 42.5±6.7 | 64.2±7.4 | 51.9±4.5 | 49.4±18.3 | 38.0±9.4 | 7.4 | 7.2 |
| LLM2Vec | BP | 40.5±6.7 | 38.5±7.0 | 48.9±10.7 | 42.7±8.6 | 54.6±9.1 | 51.5±9.6 | 25.7±14.6 | 13.4±4.1 | 9.4±1.3 | 8.8±1.1 | 29.2±6.3 | 27.8±5.5 | 34.3±4.5 | 31.7±4.2 | 53.0±9.9 | 43.8±6.6 | 52.9±12.0 | 40.9±6.5 | 61.1±7.7 | 48.7±5.6 | 50.0±20.8 | 36.5±10.4 | 8.5 | 8.7 |
| LLM2Vec | BP (appr.) | 40.3±6.6 | 38.4±7.0 | 48.6±10.0 | 42.6±8.1 | 55.2±8.0 | 52.8±8.2 | 25.0±13.7 | 13.3±3.8 | 10.1±1.5 | 10.0±1.4 | 28.0±5.8 | 26.7±5.1 | 34.6±4.6 | 31.9±4.1 | 51.7±11.2 | 42.3±7.4 | 52.1±11.3 | 42.0±6.2 | 62.5±7.2 | 49.8±4.7 | 49.7±20.2 | 36.8±10.0 | 7.4 | 7.1 |
| LLM-BP (ours) | Raw | 43.5±5.9 | 42.4±6.1 | 53.0±8.3 | 47.3±6.6 | 57.8±7.1 | **54.8±8.8** | 28.7±11.4 | 17.2±3.7 | 13.4±3.1 | 14.7±2.3 | 36.4±7.7 | 35.3±7.3 | 38.4±7.8 | 35.1±5.4 | 57.5±12.4 | 47.9±10.3 | 60.0±12.7 | 48.7±6.7 | 69.5±10.6 | 54.5±6.6 | 53.4±19.4 | 39.0±10.1 | 3.8 | 3.5 |
| LLM-BP (ours) | BP | 46.3±6.8 | 44.2±7.2 | **54.4±8.9** | **48.4±7.1** | **58.2±8.1** | 54.2±10.5 | 30.1±12.9 | 17.7±4.0 | 12.5±2.5 | 13.2±2.0 | **37.7±8.2** | **36.6±7.7** | 39.3±9.2 | 35.7±6.5 | **64.8±7.4** | **54.2±7.6** | **63.0±11.3** | **51.3±6.6** | **73.6±8.8** | **59.3±6.4** | 53.6±17.6 | 42.3±10.0 | **2.3** | **2.0** |
| LLM-BP (ours) | BP (appr.) | 44.7±6.4 | 43.5±6.4 | 53.7±8.5 | 47.9±6.8 | 57.7±7.4 | 54.5±9.2 | 29.5±12.2 | **17.7±3.9** | 13.5±3.2 | **14.8±2.4** | 36.9±7.9 | 35.8±7.5 | 39.3±8.5 | 36.0±6.1 | 61.0±10.8 | 50.8±9.7 | 62.1±11.9 | 50.5±6.5 | 71.3±9.9 | 56.6±6.3 | **54.1±19.1** | **40.8±10.6** | 2.7 | 2.4 |
| **3-Shot** | | | | | | | | | | | | | | | | | | | | | | | | | |
| SBert | Raw | 57.6±5.2 | 56.8±5.3 | 57.3±4.0 | 53.2±3.7 | 62.1±5.1 | 62.7±4.7 | 42.8±2.2 | 21.9±2.4 | 13.6±2.2 | 13.1±0.9 | 31.7±3.8 | 29.3±2.8 | 47.2±4.4 | 44.4±3.8 | 51.8±5.0 | 41.4±4.8 | 49.5±7.3 | 36.1±4.5 | 54.9±6.1 | 43.1±4.7 | 49.4±11.2 | 39.1±6.3 | 11.1 | 11.4 |
| SBert | BP | 58.7±5.3 | 57.9±5.3 | 58.6±4.5 | 54.3±4.2 | 63.7±6.1 | 64.3±5.6 | 47.0±2.4 | 23.4±2.4 | 12.5±1.9 | 11.8±0.8 | 35.1±4.4 | 32.6±3.1 | 49.5±4.5 | 46.4±4.0 | 52.2±4.7 | 42.4±4.6 | 48.9±6.9 | 36.6±4.7 | 55.0±5.9 | 43.4±4.8 | 50.2±9.5 | 39.9±5.8 | 9.6 | 9.5 |
| SBert | BP (appr.) | 58.4±5.1 | 57.6±5.2 | 57.7±4.2 | 53.6±3.9 | 62.5±5.1 | 63.1±4.7 | 43.8±2.2 | 22.3±2.5 | 13.7±2.2 | 13.3±1.0 | 32.3±3.9 | 29.9±2.8 | 48.2±4.5 | 45.4±3.9 | 51.6±4.5 | 41.5±4.6 | 49.7±7.3 | 36.4±4.4 | 54.7±5.7 | 42.9±4.2 | 49.8±10.3 | 39.4±5.9 | 10.4 | 10.5 |
| T-E-3-Large | Raw | 64.1±5.6 | 62.8±4.8 | 55.1±4.3 | 51.8±3.9 | 70.6±4.8 | 69.9±4.9 | 38.9±4.7 | 21.0±1.9 | 18.0±2.5 | 17.9±1.6 | 54.0±4.6 | 49.9±2.8 | 54.8±4.3 | 51.4±4.0 | 72.8±3.3 | 63.2±2.2 | 77.9±7.7 | 67.8±11.6 | 69.8±8.1 | 58.9±6.2 | 61.1±13.0 | 49.3±6.5 | 7.5 | 7.9 |
| T-E-3-Large | BP | **66.3±5.8** | **64.8±4.9** | 57.5±4.8 | 53.8±4.3 | **72.4±5.6** | **71.6±5.7** | 42.4±6.0 | 22.3±2.3 | 16.2±2.1 | 15.9±1.3 | 56.3±5.0 | 52.2±3.1 | 57.0±4.4 | 53.5±4.2 | 73.3±3.7 | 63.6±2.9 | 79.5±7.3 | 69.8±12.2 | 70.9±8.3 | 60.1±6.2 | 63.8±9.5 | 51.4±5.0 | 5.5 | 5.3 |
| T-E-3-Large | BP (appr.) | 65.6±5.8 | 64.1±4.9 | 56.0±4.4 | 52.6±4.0 | 71.1±5.0 | 70.4±5.1 | 40.1±5.2 | 21.5±2.0 | 18.3±2.5 | 18.2±1.6 | 54.7±5.0 | 50.6±2.9 | 56.2±4.4 | 52.8±4.1 | 73.2±3.5 | 63.6±2.9 | 78.7±7.5 | 68.8±11.8 | 70.5±8.4 | 59.4±6.3 | 62.0±12.0 | 50.0±5.8 | 6.3 | 6.5 |
| LLM2Vec | Raw | 56.9±4.9 | 56.5±4.9 | 62.3±4.2 | 58.1±3.7 | 66.6±6.4 | 66.9±5.9 | 45.4±9.2 | 24.2±3.4 | 18.1±3.4 | 17.8±1.1 | 49.5±3.6 | 46.9±2.2 | 51.7±5.1 | 49.5±5.2 | 72.6±5.4 | 61.9±6.9 | 75.0±7.1 | 71.5±8.3 | 70.7±6.1 | 59.5±4.7 | 63.3±10.2 | 49.6±3.9 | 8.2 | 8.0 |
| LLM2Vec | BP | 59.6±5.3 | 59.0±5.3 | 64.6±4.5 | 59.9±3.9 | 67.3±7.5 | 67.4±6.7 | 48.9±10.5 | 25.8±4.0 | 16.4±2.8 | 15.6±1.0 | 52.8±3.9 | 50.1±2.5 | 53.9±6.0 | 51.5±5.7 | 74.1±5.2 | 64.2±6.5 | 79.6±4.6 | 74.5±8.1 | 72.6±5.0 | 60.7±4.0 | 65.0±10.0 | 51.7±4.1 | 5.5 | 5.5 |
| LLM2Vec | BP (appr.) | 59.2±5.0 | 58.7±4.9 | 63.7±4.3 | 59.2±3.7 | 67.1±6.6 | 67.4±6.6 | 47.4±9.8 | 25.4±3.7 | 18.7±3.8 | 18.3±1.4 | 50.9±3.7 | 48.2±2.3 | 54.0±5.5 | 51.5±5.3 | 74.0±6.4 | 64.0±6.4 | 77.6±5.2 | 73.2±7.4 | 72.2±5.6 | 60.6±6.4 | 64.6±10.1 | 50.8±4.1 | 6.1 | 5.9 |
| LLM-BP (ours) | Raw | 60.0±4.0 | 59.2±3.9 | 64.6±4.4 | 60.3±3.6 | 68.7±5.2 | 69.0±4.8 | 59.8±6.0 | 32.2±3.0 | 22.3±3.3 | 22.3±1.5 | 58.2±4.0 | 55.7±2.6 | 55.1±4.9 | 53.5±4.6 | 79.9±3.8 | 72.2±5.7 | 82.5±5.4 | 79.0±6.2 | 83.0±4.0 | 72.3±4.6 | 71.4±12.5 | 58.3±6.0 | 3.6 | 3.5 |
| LLM-BP (ours) | BP | 64.3±4.6 | 63.1±4.3 | **66.5±4.8** | **61.8±4.0** | 69.7±6.2 | 69.7±5.8 | **62.8±6.1** | 33.4±3.2 | 20.3±2.9 | 19.6±1.4 | **60.2±4.7** | **57.8±2.8** | **57.8±5.6** | **56.0±5.2** | 81.1±4.0 | 73.6±5.5 | **85.4±3.5** | **85.4±3.5** | 84.0±4.0 | 73.3±4.8 | 71.9±10.6 | **60.1±5.7** | **1.9** | **1.7** |
| LLM-BP (ours) | BP (appr.) | 61.7±4.2 | 60.9±4.0 | 65.7±4.8 | 61.3±3.9 | 69.2±5.5 | 69.5±5.1 | 61.6±5.9 | 33.2±3.0 | **22.7±3.5** | **22.7±1.6** | 59.2±4.1 | 56.7±2.7 | 57.4±4.7 | 55.8±4.5 | **81.3±4.0** | 73.6±5.7 | 84.6±4.7 | 81.3±6.0 | **84.3±4.1** | 73.2±5.2 | **72.1±11.8** | 59.1±6.2 | 2.2 | 2.4 |
| **5-Shot** | | | | | | | | | | | | | | | | | | | | | | | | | |
| SBert | Raw | 61.4±3.5 | 60.8±3.2 | 61.4±4.6 | 57.4±4.2 | 66.5±4.8 | 67.3±4.3 | 46.7±4.5 | 25.3±2.5 | 16.3±2.3 | 15.7±0.9 | 39.1±3.1 | 36.0±1.8 | 50.5±2.5 | 48.8±2.1 | 55.9±3.6 | 45.4±3.7 | 56.1±6.9 | 40.5±5.7 | 59.8±4.5 | 45.5±4.3 | 56.1±7.6 | 44.1±3.2 | 11.5 | 11.6 |
| SBert | BP | 62.7±4.0 | 62.0±3.7 | 62.6±4.7 | 58.3±4.1 | 67.9±5.7 | 68.7±4.9 | 50.8±5.2 | 26.9±2.6 | 14.7±1.9 | 13.9±0.8 | 43.4±3.6 | 40.2±2.1 | 52.3±6.0 | 49.9±2.0 | 57.2±3.6 | 46.4±3.7 | 55.8±7.0 | 41.6±5.5 | 60.0±4.0 | 45.5±3.5 | 53.8±6.1 | 43.1±2.5 | 10.2 | 10.2 |
| SBert | BP (appr.) | 62.2±3.7 | 61.5±3.4 | 61.8±4.6 | 57.8±4.2 | 67.0±4.8 | 67.8±4.3 | 47.8±4.7 | 25.9±2.5 | 16.6±2.4 | 16.0±0.9 | 40.0±3.3 | 36.9±1.9 | 51.6±2.5 | 49.9±2.0 | 55.9±3.5 | 45.5±3.8 | 56.5±7.0 | 41.2±5.8 | 59.9±4.1 | 45.7±4.0 | 55.1±7.0 | 43.7±2.9 | 10.6 | 10.6 |
| T-E-3-Large | Raw | 69.4±3.1 | 68.4±2.7 | 60.9±4.4 | 57.3±3.6 | 74.8±5.0 | 74.4±4.9 | 53.4±5.7 | 26.9±2.9 | 21.7±3.0 | 22.2±1.1 | 61.7±4.9 | 57.7±3.0 | 63.9±5.4 | 59.6±2.6 | 62.0±3.3 | 59.1±3.6 | 75.3±4.1 | 67.7±3.3 | 84.2±3.0 | 81.2±2.7 | 76.3±7.2 | 63.3±6.5 | 5.4 | 5.5 |
| T-E-3-Large | BP | **70.6±3.1** | **69.5±2.7** | 63.3±4.6 | 59.3±3.8 | **75.6±5.5** | **75.0±5.6** | 53.4±5.7 | 28.3±3.1 | 19.5±2.3 | 19.4±0.9 | 63.9±5.9 | 59.6±2.6 | 62.0±3.3 | 59.1±3.6 | 75.3±4.1 | 67.9±3.1 | 84.7±2.9 | 80.6±6.2 | 76.3±7.2 | 63.3±6.5 | 65.4±9.1 | 56.2±4.6 | 5.4 | 5.5 |
| T-E-3-Large | BP (appr.) | 70.5±3.4 | 69.4±2.9 | 61.9±4.4 | 58.1±3.7 | 75.4±5.2 | 74.9±5.2 | 53.5±5.5 | 27.6±3.1 | 22.1±3.1 | 22.4±1.2 | 62.5±5.1 | 58.1±2.5 | 61.0±3.9 | 58.3±3.5 | 75.4±4.2 | 68.3±3.1 | 84.2±2.9 | 81.3±2.6 | 75.6±7.6 | 62.7±7.6 | 64.1±9.7 | 54.8±4.7 | 6.1 | 5.8 |
| LLM2Vec | Raw | 61.6±3.6 | 61.1±3.4 | 64.6±5.0 | 61.0±4.5 | 71.4±5.9 | 71.8±5.4 | 54.4±7.0 | 29.8±2.9 | 22.5±2.9 | 22.1±1.0 | 56.8±4.5 | 54.4±2.1 | 58.5±3.6 | 56.4±4.2 | 75.5±5.6 | 68.3±5.1 | 81.8±3.8 | 77.6±2.7 | 78.4±5.0 | 64.3±4.4 | 70.3±8.0 | 55.6±3.6 | 7.5 | 7.7 |
| LLM2Vec | BP | 64.0±3.4 | 63.3±3.4 | 66.9±5.3 | 62.9±4.7 | 72.9±6.8 | 73.2±6.2 | 58.5±7.4 | 31.3±3.0 | 20.3±2.5 | 19.1±0.8 | 58.3±4.8 | 55.7±2.5 | 61.7±3.9 | 59.1±4.4 | 76.2±6.0 | 69.2±6.0 | 83.8±2.5 | 80.5±2.1 | 81.0±4.3 | 67.0±3.9 | 71.1±8.3 | 57.7±4.2 | 5.5 | 5.3 |
| LLM2Vec | BP (appr.) | 64.0±3.5 | 63.3±3.5 | 65.9±5.1 | 62.0±4.6 | 72.0±6.1 | 72.4±5.6 | 56.7±7.1 | 31.1±3.1 | 23.2±3.1 | 22.7±1.0 | 58.3±4.8 | 55.9±2.2 | 61.0±3.9 | 58.9±4.4 | 76.3±6.0 | 69.1±5.8 | 83.0±3.4 | 79.4±2.3 | 80.6±4.8 | 66.6±4.4 | 70.7±8.5 | 56.5±4.0 | 5.9 | 6.0 |
| LLM-BP (ours) | Raw | 65.1±2.8 | 64.4±3.0 | 65.5±5.4 | 61.9±4.7 | 73.6±6.2 | 73.9±5.9 | 46.5±5.6 | 25.6±2.5 | 26.7±1.1 | 26.1±0.7 | 54.3±5.4 | 61.3±2.2 | 62.2±3.6 | 60.4±3.5 | 76.5±4.8 | 87.3±3.4 | 85.9±2.6 | 87.3±3.4 | 76.1±4.9 | **77.5±8.8** | 63.5±4.1 | | 3.5 | 3.6 |
| LLM-BP (ours) | BP | 69.5±3 | 68.3±3.3 | **67.4±5.8** | **63.3±5.1** | 74.0±6.5 | 74.1±6.2 | **68.4±4.8** | **38.7±3.7** | 24.3±2.2 | 23.5±0.9 | **66.6±5.6** | **63.6±2.4** | **65.2±3.7** | **63.1±3.7** | 83.3±3.1 | 77.6±4.8 | **88.2±2.8** | 86.3±2.6 | **88.4±2.5** | **76.8±4.0** | 75.8±7.7 | 64.0±4.7 | **1.9** | **1.9** |
| LLM-BP (ours) | BP (appr.) | 67.0±2.7 | 66.0±2.8 | 67.5±5.8 | 62.8±5.0 | 73.9±6.2 | 74.1±5.9 | 66.9±4.7 | 38.4±3.6 | **26.9±2.7** | **27.0±1.1** | 65.5±5.5 | 62.5±2.4 | 64.6±3.4 | 62.3±3.5 | **83.6±2.9** | 77.6±4.4 | 87.9±3.7 | **86.4±2.6** | 88.4±2.8 | 76.6±4.9 | 77.4±9.0 | **64.8±5.4** | 2.4 | 2.4 |
| **10-shot** | | | | | | | | | | | | | | | | | | | | | | | | | |
| SBert | Raw | 69.8±3.3 | 68.5±3.1 | 67.0±2.0 | 63.0±1.8 | 68.9±4.1 | 69.3±4.6 | 57.3±2.7 | 31.2±0.9 | 20.2±1.8 | 19.7±0.9 | 46.5±3.2 | 45.0±1.7 | 58.0±1.1 | 58.1±1.8 | 61.7±2.9 | 48.1±4.6 | 60.5±3.7 | 47.2±6.6 | 66.1±3.4 | 49.5±4.0 | 62.8±4.1 | 44.1±3.2 | 11.3 | 11.2 |
| SBert | BP | 70.6±3.2 | 69.2±2.9 | 68.3±1.9 | 64.1±1.7 | 70.5±4.9 | 70.8±5.5 | 61.6±2.9 | 33.2±1.2 | 18.0±1.4 | 17.3±0.8 | 51.6±3.9 | 48.3±2.3 | 62.4±2.2 | 60.8±1.9 | 62.0±2.7 | 48.8±4.7 | 60.0±3.3 | 45.5±4.8 | 66.2±4.0 | 49.4±3.8 | 58.5±4.3 | 45.3±3.7 | 9.8 | 10.1 |
| SBert | BP (appr.) | 70.3±3.2 | 68.9±3.1 | 67.5±1.8 | 63.4±1.7 | 69.3±4.3 | 69.7±4.8 | 58.5±2.8 | 31.9±1.0 | 20.6±1.9 | 20.1±0.9 | 47.8±3.4 | 44.6±2.0 | 60.2±2.5 | 59.5±1.8 | 62.0±2.6 | 48.7±4.6 | 60.4±3.6 | 46.6±6.2 | 66.4±3.9 | 49.6±4.0 | 60.3±4.6 | 45.7±3.5 | 10.4 | 10.5 |
| T-E-3-Large | Raw | 74.9±1.2 | 73.7±1.4 | 65.7±1.8 | 61.9±1.5 | 78.3±4.2 | 78.1±4.0 | 59.8±6.5 | 33.0±2.1 | 27.1±2.3 | 26.9±1.4 | 68.1±3.8 | 63.1±2.1 | 68.8±1.9 | 66.6±1.7 | 80.1±2.1 | 71.6±3.4 | 86.5±3.2 | 84.2±2.9 | 81.8±3.5 | 68.3±6.5 | 74.7±4.2 | 60.0±4.6 | 7.5 | 7.6 |
| T-E-3-Large | BP | **76.6±1.3** | **75.2±1.5** | 68.3±2.0 | 64.0±1.7 | **79.1±4.4** | **78.7±4.3** | 63.9±6.7 | 34.8±2.2 | 24.0±1.7 | 23.4±1.0 | 70.0±4.1 | 65.2±2.1 | 71.3±2.0 | 68.8±1.8 | 80.5±2.7 | 72.3±4.1 | 87.2±2.6 | 85.4±2.8 | 81.9±2.9 | 68.7±6.5 | 73.7±3.9 | 60.7±3.7 | 5.5 | 5.5 |
| T-E-3-Large | BP (appr.) | 76.4±1.3 | 75.0±1.6 | 66.7±1.9 | 62.7±1.6 | 78.7±4.3 | 78.4±4.2 | 61.3±6.7 | 33.9±2.2 | 27.6±2.4 | 27.3±1.4 | 68.8±3.9 | 63.9±2.1 | 70.2±1.9 | 68.0±1.8 | 80.7±1.7 | 72.2±3.1 | 86.7±3.3 | 84.6±3.2 | 82.0±3.5 | 68.7±6.5 | 74.3±4.2 | 60.0±4.2 | 6.1 | 6.2 |
| LLM2Vec | Raw | 68.1±2.7 | 66.9±3.0 | 68.0±2.8 | 64.5±2.6 | 74.2±3.7 | 74.5±3.4 | 59.6±4.2 | 37.5±1.8 | 27.2±2.8 | 27.0±1.3 | 63.4±3.3 | 60.7±1.6 | 68.8±3.4 | 66.7±2.4 | 80.6±3.6 | 72.9±4.4 | 85.9±3.0 | 82.6±4.4 | 84.5±1.7 | 69.0±5.1 | 81.6±4.5 | 61.4±5.9 | 7.8 | 7.5 |
| LLM2Vec | BP | 70.4±3.0 | 69.3±3.2 | 70.6±2.9 | 66.5±2.7 | 74.9±4.1 | 75.0±3.8 | 69.9±4.2 | 39.6±2.4 | 24.6±2.5 | 23.2±1.1 | 66.2±3.6 | 63.6±1.7 | 71.2±2.6 | 68.8±2.4 | 80.9±3.1 | 73.5±4.2 | 87.3±2.0 | 84.9±2.8 | 85.9±2.4 | 70.3±4.3 | 80.5±4.4 | 63.0±5.3 | 5.3 | 5.2 |
| LLM2Vec | BP (appr.) | 70.0±3.0 | 68.8±3.1 | 69.5±2.8 | 65.7±2.6 | 73.5±4.2 | 73.8±4.0 | 68.2±4.2 | 39.5±2.1 | 28.1±3.1 | 27.8±1.3 | 64.9±3.5 | 62.2±1.6 | 70.1±2.4 | 67.6±2.1 | 81.5±2.9 | 73.9±3.7 | 86.6±2.6 | 83.8±3.3 | 86.0±1.8 | 70.5±4.6 | 82.1±4.8 | 64.6±6.0 | 5.7 | 5.4 |
| LLM-BP (ours) | Raw | 70.4±3.0 | 69.1±3.4 | 69.1±2.6 | 65.5±2.2 | 74.3±4.3 | 74.5±4.2 | 72.3±3.1 | 43.5±2.1 | 30.8±2.0 | 30.4±0.9 | 68.5±3.4 | 65.2±1.4 | 71.4±2.3 | 69.4±2.3 | 85.9±3.2 | 80.0±4.2 | 90.7±2.2 | 88.0±3.6 | 90.7±1.7 | 78.7±5.3 | 85.9±2.7 | 68.8±3.7 | 3.6 | 3.6 |
| LLM-BP (ours) | BP | **74.2±3.1** | **72.5±3.2** | **71.3±2.8** | **67.3±2.2** | 75.5±4.2 | 75.6±4.0 | **74.5±2.9** | **44.7±2.4** | 28.2±1.8 | 26.7±0.8 | **70.5±3.5** | **67.4±1.6** | **73.7±2.4** | 71.4±2.4 | 86.1±3.1 | 80.8±4.1 | **90.7±1.8** | 88.1±3.1 | 90.6±1.7 | 76.9±5.2 | 83.1±3.5 | 67.9±3.5 | **2.4** | 2.6 |
| LLM-BP (ours) | BP (appr.) | 71.9±3.5 | 70.6±3.7 | 70.4±2.6 | 66.7±2.1 | 74.8±4.4 | 75.0±4.2 | 73.6±3.0 | 44.6±2.4 | **31.4±2.1** | **30.8±1.0** | 69.7±3.5 | 66.5±1.5 | 73.6±2.2 | **71.6±2.1** | **86.8±2.9** | 81.0±3.8 | 90.7±1.9 | **88.1±3.0** | **91.7±1.7** | **78.8±5.2** | 85.0±3.3 | 68.3±4.0 | 2.5 | **2.3** |

Table 10: Few-Shot Performance. 'T-E-3-Large' is short for Text-Embedding-3-Large.

We leave the design of task-adaptive embeddings and generalized graph structural utilization for link prediction as future work, including task-adaptive encoding prompts.

## E. Prompts

### E.1. Task Description for Vanilla LLM2Vec without Class Information

Table. 12 shows the task description for vanilla LLM2Vec encoder across all the datasets.

### E.2. Prompts for Vanilla LLMs

Table. 13 shows tha task description for vanilla LLM decoders.

| | Citation Graph | | | E-Commerce & Knowledge Graph | | | |
| --- | --- | --- | --- | --- | --- | --- | --- |
| | Cora | Citeseer | Pubmed | History | Children | Sportsfit | Wikics |
| OFA | 0.492 | – | 0.481 | 0.431 | 0.484 | 0.517 | – |
| LLaGA | 0.527 | – | 0.543 | 0.515 | 0.500 | 0.502 | – |
| GraphGPT | 0.520 | – | 0.569 | 0.449 | 0.422 | 0.597 | – |
| TEA-GLM | 0.586 | – | 0.689 | 0.579 | 0.571 | 0.553 | – |
| SBert | 0.979±0.033 | 0.990±0.001 | 0.979±0.003 | 0.985±0.002 | 0.972±0.030 | 0.975±0.003 | 0.972±0.003 |
| Text-Embedding-3-Large | 0.975±0.003 | 0.989±0.002 | 0.979±0.003 | 0.985±0.001 | 0.980±0.002 | 0.987±0.001 | 0.977±0.002 |
| LLM2Vec | 0.966±0.004 | 0.982±0.002 | 0.970±0.003 | 0.971±0.002 | 0.973±0.003 | 0.975±0.002 | 0.978±0.003 |

Table 11: Performance on zero-shot link prediction tasks (AUC). Results of baselines are from (Wang et al., 2024).

| Dataset | Task Description |
| --- | --- |
| Cora | Encode the text of machine learning papers: |
| Citeseer | Encode the description or opening text of scientific publications: |
| Pubmed | Encode the title and abstract of scientific publications: |
| History | Encode the description or title of the book: |
| Children | Encode the description or title of the child literature: |
| Sportsfit | Encode the title of a good in sports & fitness: |
| Wikics | Encode the entry and content of wikipedia: |
| Cornell | Encode the webpage text: |
| Texas | Encode the webpage text: |
| Wisconsin | Encode the webpage text: |
| Washington | Encode the webpage text: |

Table 12: {Task description} for vanilla LLM2Vec (Li et al., 2024a) encoder. See Eq. 8 for detailed prompting format.

| Dataset | Task Description |
| --- | --- |
| Cora | opening text of machine learning papers |
| Citeseer | description or opening text of scientific publications |
| Pubmed | title and abstract of scientific publications |
| History | description or title of the book |
| Children | description or title of the child literature |
| Sportsfit | the title of a good in sports & fitness |
| Wikics | entry and content of wikipedia |
| Cornell | webpage text |
| Texas | webpage text |
| Wisconsin | webpage text |
| Washington | webpage text |

Table 13: {Task description} in the prompts for both vanilla LLM decoders (See Section. B.3) and task-adaptive encoder (See Section. B.2).

