# OpenReview forum: "Generalization Principles for Inference over Text-Attributed Graphs  with Large Language Models"
_ICML.cc/2025/Conference — ICML 2025 poster_

### Official Review · Reviewer_gfhg · 2025-03-11

**Overall Recommendation:** 3

**Summary:**

This paper introduces LLM-BP, a framework for zero-shot inference on text-attributed graphs using large language models. The framework requires no training and generalizes well across homophilic and heterophilic graphs. Experiments show that LLM-BP outperforms existing methods.

**Claims And Evidence:**

Yes.

**Essential References Not Discussed:**

None.

**Experimental Designs Or Analyses:**

- Error bars or variance metrics (e.g., standard deviation) are missing in the main results (Table 2).
- Sensitivity of results to the number of sampled edges is not analyzed.
- Experiments on computational overhead need to be supplemented.

**Methods And Evaluation Criteria:**

Yes.

**Other Comments Or Suggestions:**

None.

**Other Strengths And Weaknesses:**

None.

**Questions For Authors:**

I am quite concerned about the issue of computational overhead. If we select two nodes each time to query the LLM, the cost for large graph datasets will be extremely high. Can the authors help explain this issue?

**Relation To Broader Scientific Literature:**

The work situates itself within the growing literature on integrating LLMs with graph learning.

**Theoretical Claims:**

While the derivation of BP updates (Section 3.4, Appendix C) is logically presented, the theoretical guarantees for convergence or approximation bounds of the proposed BP variant (Eq. 7) are not rigorously analyzed.

---

> ### Author Rebuttal · Authors · 2025-04-01
>
> We thank *reviewer gfhg*’s time in reviewing our paper and their constructive comments. Below we try to address the concerns.
>
> **1. [Convergence of BP]**
>
> We acknowledge that there is no general convergence guarantee for BP on graphs with loops. However, LLM-BP does not require BP or its approximate variant (BP appr.) to converge. The purpose of incorporating BP (or BP appr.) is to demonstrate the principled use of neighbor information. Even without full convergence, each BP iteration corresponds to a single round of statistical inference based on neighbors and can still yield meaningful performance gains. As shown in our experiments, running BP to full convergence is not necessary to achieve strong predictive performance. A few iterations are often sufficient to realize the benefits of structured information propagation.
>
> **2. [Variance metric in Table. 2]**
>
> Below, we report the results of a significance test evaluating the improvements from LLM-BP. Each method was run 100 times per dataset with 100 different random seeds. The table presents the estimated lower/upper bounds of performance improvement under 90% confidence interval, along with the $p$-values for statistical significance. Comparisons are listed in the first column.
>
> As highlighted in bolded texts, task-adaptive encoding yields statistically significant improvement over LLM2Vec on 9 out of 11 datasets, and outperforms Text-Embedding-3-Large on 8 out of 11. Furthermore, LLM-BP provides statistically significant gains over task-adaptive encoding on 10 out of 11 datasets, while LLM-BP (appr.) achieves improvement on all 11 datasets.
>
> |  |  | Cora | Citeseer | Pubmed | History | Child | Sportsfit | Wikics | Cornell | Texas | Wisc | Wash  |
> |:---:|:---:|:---:|:---:|:---:|:---:|:---:|:---:|:---:|:---:|:---:|:---:|:---:|
> | Task-adaptive encoding vs. Text-Embedidng-3-Large| Low,Up | -0.3,-0.2 | 0.5,1.0 | -0.3,1.0 | 6.9,9.1 | 4.1,4.4 | 0.7, .8 | 1.1,2.0 | 0.3,0.4 | 3.1,4.7 | 0.1,1.2 | -0.0,-0.1  |
> | | P | 6e-27 | **8e-7** | 0.51 | **7e-21** | **5e-59** | **0.03** | **1e-9** | **2e-25** | **1e-9** | **0.07** | 1e-13  |
> | Task-adaptive encoding vs. LLM2Vec| Low,Up | 0.7,1.3 | -0.2,1.3 | 1.3,2.3 | 3.2,5.2 | 3.3,3.7 | 1.0,2.6 | 1.0,1.8 | 0.4,0.5 | 1.5,2.7 | 0.1,0.8 | -0.2,0.5  |
> | | P | **1e-8** | 0.38 | **1e-10** | **1e-9** | **3e-59** | **1e-4** | **3e-8** | **7e-18** | **5e-8** | **1e-3** | 0.82  |
> | LLM-BP vs. Task-adaptive Encoding | Low,Up | 3.6,4.0 | 2.3,2.5 | 0.6,1.1 | 1.4,1.7 | -0.3,0.4 | 2.3,2.5 | 4.1,4.4 | 0.3,0.4 | 3.2,3.7 | 2.0,2.6 | 5.5,6.5  |
> | | P | **2e-52** | **1e-15** | **1e-6** | **6e-35** | 0.26 | **1e-69** | **8e-65** | **1e-15** | **3e-40** | **1e-16** | **1e-35**  |
> | LLM-BP (appr.) vs. Task-adaptive Encoding | Low,Up | 2.6, 2.8 | 1.5,1.6 | 1.6,1.8 | 1.1,1.3 | 0.3,0.5 | 2.0,2.1 | 4.4,4.8 | 1.9,2.3 | 1.0,1.5 | 0.4,0.6 | 2.8,3.5  |
> | | P | **1e-52** | **2e-57** | **3e-9** | **1e-43** | **6e-20** | **3e-18** | **4e-60** | **2e-4** | **5e-14** | **0.06** | **3e-27** |
>
> **2. [Sensitivity Analysis]**
>
> Thanks for the advice in sensitivity analysis on the edge number in predicting homophily ratio. The results of prediction by GPT-4o-mini and ground truth are as follows, which exhibits a stable prediction with the sampled edge number from 40 to100:
>
> |  | Cora |  | Citeseer |  | Pubmed |  | Bookhis |  | Bookchild |  | Sportsfit |  | Wikics |   |
> |:---:|:---:|:---:|:---:|:---:|:---:|:---:|:---:|:---:|:---:|:---:|:---:|:---:|:---:|:---:|
> |  | value | gap | value | gap | value | gap | value | gap | value | gap | value | gap | value | gap  |
> | ground truth | 0.81 | - | 0.76 | - | 0.79 | - | 0.66 | - | 0.46 | - | 0.90 | - | 0.67 | -  |
> | 100 | 0.70 | 0.11 | 0.81 | 0.05 | 0.81 | 0.02 | 0.73 | 0.07 | 0.35 | 0.11 | 0.81 | 0.09 | 0.52 | 0.15  |
> | 80 | 0.70 | 0.11 | 0.77 | 0.01 | 0.83 | 0.04 | 0.75 | 0.09 | 0.37 | 0.09 | 0.76 | 0.14 | 0.55 | 0.12  |
> | 40 | 0.65 | 0.16 | 0.77 | 0.01 | 0.81 | 0.02 | 0.75 | 0.09 | 0.33 | 0.13 | 0.75 | 0.15 | 0.50 | 0.17  |
>
>
> **3. [Computational Overhead Analysis]**
>
> It is unclear to us why querying two nodes simultaneously would introduce significant computational overhead. To clarify, the complexity of LLM-BP consists of two main components:
>
> First, obtaining node or class embeddings using an encoder has the same order of complexity as the baseline methods.
>
> Second, the BP (or appr.) step can be viewed as a non-parametric GNN. Therefore, its computational complexity is comparable to that of standard GNN-based baselines. Moreover, in contrast to methods involving graph adaptors coupled with LLM decoders, LLM-BP is more time-efficient, as it does not rely on the costly decoding process of LLMs.
>
> If there are specific concerns regarding computational complexity that we may have overlooked, we would greatly appreciate further clarification.
>
> ---
>
> We hope that we have addressed the concerns of *reviewer gfhg*, we would be happy to respond to any other questions.

---

### Official Review · Reviewer_JRhf · 2025-03-12

**Overall Recommendation:** 4

**Summary:**

The paper proposes a new method to enhance LLM's ability on graph learning tasks. It first proposes to incorporate task and class information into the node embedding generated by the language model, it then proposes to use belief propagation on pseudo-labels of the nodes to enhance prediction. Experiments show consistent improvement over existing methods.

## update after rebuttal

After the rebuttal phase, the paper still stands out as a very interesting paper with outstanding performance, despite some misleading discussion on the contribution, which should be easy to fix during revision. Hence, I am keeping my score.

**Claims And Evidence:**

While the author claims principal 1 as a novel observation, incorporating task information to generate embedding has been widely discussed by many works that the author also mentioned. The connection to them should be emphasized.

**Essential References Not Discussed:**

NA

**Experimental Designs Or Analyses:**

How do you conduct zero-shot experiment with sBERT? Do you also compare the cos similarity? In particular, Figure 2 shows that sBERT is better than LLaGA, what's the setup for this comparison.

**Methods And Evaluation Criteria:**

Yes, very solid evaluation.

**Other Comments Or Suggestions:**

Is $q_k^c$ in equation 5 $h_k^c$, or is it the candidate embedding introduced earlier, or are they the same thing?

**Other Strengths And Weaknesses:**

The paper is well-written, and backed by a comprehensive list of experiments.

**Questions For Authors:**

I am not entirely sure how the model handles heterophilic cases, or is advantageous in heterophily at all. From Figure 6, we see that using the LLM embedding, the heterophlic performance is already good.

**Relation To Broader Scientific Literature:**

The paper's contribution is mainly the pseudo-label belief propagation process powered by LLM, which seem to bring significant performance improvement. The enhanced embedding paradigm is also effective, yet the improvement on this end is not surprising, and has been studied in various related works.

**Theoretical Claims:**

NA.

---

> ### Author Rebuttal · Authors · 2025-04-01
>
> We sincerely thank the time and efforts *reviewer JRhf* took to review our paper. *Reviewer JRhf* provides some insightful questions and constructive suggestions to further improve the paper’s quality. Below we try our best to address the concerns:
>
> **1. [Connections with other works]**
>
> We agree that in-context encoding has been shown in prior work (e.g., [1]) to improve text embedding quality, and we will revise our manuscript to strengthen the discussion of these connections. However, to the best of our knowledge, no existing work has explored or claimed that incorporating class information improves embedding quality in classification tasks, let alone in the graph domain. We therefore consider this a novel contribution of our work. That said, we would greatly appreciate it if the reviewer could point us to any relevant studies we may have missed, and we would be glad to further discuss and cite them in the revised paper.
>
> **2. [Zero-shot experiment setting with SBert]**
>
> The reviewer's understanding is correct. The evaluation of SBert is as follows: 1) Obtain the class embeddings and node embeddings with SBert 2) directly compare the cosine similarity between node embeddings and class embeddings, assign it to the class with highest similarity score. Note that this process does not involve any graph structure usage, and it still could achieve better zero-shot performance than the LLMs-with-Graph-Adaptor baselines that require alignment.
>
> **3. [LLM-BP’s advantages on heterophilic data]**
>
> Thank you for the comment. We offer two clarifications in response:
>
> First, on certain heterophilic datasets such as *Cornell*, the quality of LLM-generated node embeddings is indeed high, which can yield strong performance even without leveraging graph structure. However, this is not always the case. For other heterophilic graphs (*Texas* and *Washington*) where the node embedding quality is comparatively lower, relying solely on LLM embeddings proves insufficient. In these cases, the BP algorithm significantly improves classification accuracy.
>
> Second, we acknowledge that the performance of class embeddings on heterophilic graphs may have been somewhat overestimated in our original setup. Because heterophilic datasets typically have fewer nodes, sampling 20× the number of classes (as we did for larger homophilic graphs) to derive class embeddings, covers nearly a third of all nodes in heterophilic graphs. This made the class embeddings overly similar to the mean of node embeddings, thus obscuring the differences between baselines.
>
> In updated experiments, we revised the sampling ratio to 3× and 5× the number of classes for these smaller graphs. Under this adjusted setup, the improvements provided by LLM-BP, through both task-adaptive encoding and the BP algorithm, become much more pronounced, reinforcing its advantage in heterophilic settings.
>
> |  | Cornell |  |  |  | Texas |  |  |  | Wisc |  |  |  | Wash |  |  |  |
> |:---:|:---:|:---:|:---:|:---:|:---:|:---:|:---:|:---:|:---:|:---:|:---:|:---:|:---:|:---:|:---:|:---:|
> | # samples for class embedding | 3c |  | 5c |  | 3c |  | 5c |  | 3c |  | 5c |  | 3c |  | 5c |  |
> |  | Acc | F1 | Acc | F1 | Acc | F1 | Acc | F1 | Acc | F1 | Acc | F1 | Acc | F1 | Acc | F1 |
> | Text-Embedding-3-Large | 51.65 | 40.50 | 57.78 | 45.35 | 63.72 | 53.03 | 64.90 | 49.80 | 56.53 | 50.92 | 59.38 | 51.61 | 56.28 | 42.09 | 61.06 | 45.21 |
> | LLM2Vec | 45.28 | 35.74 | 51.20 | 38.35 | 59.17 | 48.23 | 62.11 | 46.40 | 53.78 | 46.79 | 57.69 | 48.69 | 51.39 | 37.74 | 56.59 | 39.31 |
> | Task-Adaptive Encoding | 52.68 | 40.40 | 60.71 | 47.40 | 64.85 | 51.99 | 67.90 | 52.60 | 60.31 | 52.35 | 64.26 | 53.72 | 54.94 | 41.33 | 60.82 | 45.10 |
> | LLM-BP | 56.57 | 44.05 | 66.05 | 50.35 | 66.99 | 53.97 | 69.47 | 53.70 | 61.04 | 53.25 | 65.72 | 54.29 | 55.41 | 43.65 | 62.80 | 46.25 |
> | LLM-BP (appr.) | 56.91 | 43.60 | 65.67 | 50.80 | 67.12 | 53.98 | 70.48 | 55.18 | 59.70 | 52.50 | 63.32 | 52.02 | 55.64 | 43.00 | 62.12 | 46.69 |
>
>
> **4. [Typo in Equation 5.]**
>
> Thanks for pointing out the typo, the $q^C_k$ here in Equation. (5) should be fixed with $h^C_k$.
>
> [1] Making text embedders few-shot learners. Chaofan Li, MingHao Qin, Shitao Xiao, Jianlyu Chen, Kun Luo, Yingxia Shao, Defu Lian, Zheng Liu. ICLR 2025.

---

> > ### Comment · Reviewer_JRhf · 2025-04-03
> >
> > On pincipal one: from broader research, providing a set of candidate classes as context for LLM to predict is common practice; from general graph models, some work the author mentioned, like GOFA, also include class information on nodes. The point here is that, while the exact approach to inject class information might not be present, this is essentially using class information as context, which is not new to the literature.
> >
> > The authors addressed all my other questions.

---

> > > ### Author Response · Authors · 2025-04-08
> > >
> > > Thank you for your response. We appreciate the insights you provided, and we will ensure a more thorough discussion of the relevant work on task-adaptive embedding in the camera-ready version of our paper. We're also glad your other concerns have been satisfactorily addressed.

---

### Official Review · Reviewer_ppW1 · 2025-03-14

**Overall Recommendation:** 1

**Summary:**

This paper tackles node classification on text-attributed graphs (TAGs) -- graphs where each node has a textual description but labelled examples are scarce. It identifies two major challenges of existing approaches that utilise LLMs for this task: *(i)* LLMs have limited context length, making it hard to include extensive neighbor information for a node, and *(ii)* there is a mismatch between typical node embeddings (from graph encoders) and the token-based input space of LLMs. To address these, the authors propose LLM-BP, a framework based on two principles. First, they create task-adaptive text embeddings for each node by leveraging the idea of LLM2Vec with carefully crafted prompts that include task description and class information. Second, instead of feeding aggregated neighbor embeddings into an LLM, they perform a belief propagation (BP)-inspired label inference on the graph, using an LLM to estimate the edge coupling parameters (essentially the graph’s homophily/heterophily) for adaptive neighbour aggregation.

**Claims And Evidence:**

Several claims regarding LLM-BP’s effectiveness are questionable, particularly in how its performance is measured and presented.

1. **Misleading Use of Average Ranking**: The claim that "LLM-BP and LLM-BP (appr.) achieve the highest average ranking" is misleading due to dataset size imbalance. Homophilic datasets (e.g., Cora, Citeseer, Pubmed) are much larger than heterophilic ones (e.g., Cornell, Texas, Wisconsin), yet the ranking metric treats all datasets equally. This underemphasises LLM-BP’s weaker performance on large datasets while inflating its success on smaller ones, distorting the overall conclusion.
2. **Limited Gains on Large, Homophilic Graphs**: LLM-BP performs similarly to or worse than GPT-4o on homophilic graphs, where node text alone is highly informative and graph structure contributes little additional value. Since homophilic datasets are much larger, this suggests that the ranking metric overstates LLM-BP’s generalisation ability.
3. **Misleading "Pre-training-Free" Claim**: The paper states: *"Unlike LLM-BP which is training-free, most of the baselines–except from vanilla encoders, LLMs or NA–require pre-training."* This is misleading because LLM-BP relies on a pre-trained LLM, just as other methods rely on pre-trained models (e.g., fine-tuned graph encoders). The correct distinction is that LLM-BP does not require fine-tuning on graph-specific data.

The third point is relatively minor but the first two demonstrate strong limitations of this work.

**Essential References Not Discussed:**

Not found.

**Experimental Designs Or Analyses:**

This work has a good coverage of datasets and baselines. Issues regarding the analyses have been discussed above.

**Methods And Evaluation Criteria:**

The general applicability of the proposed methods remains questionable and it has been discussed in *Claims And Evidence*.

**Other Comments Or Suggestions:**

- Formatting issue: The main paper, which does not include a conclusion, exceeds the page limit a bit.

**Other Strengths And Weaknesses:**

n/a

**Questions For Authors:**

n/a

**Relation To Broader Scientific Literature:**

This work is situated at the intersection of graph machine learning and large language models, and the authors do a commendable job relating it to prior research.

**Theoretical Claims:**

The proposed methodology is largely based on established concepts in probabilistic graph models and appear to be correctly applied. There are no entirely new theoretical claims that require deep proof.

---

> ### Author Rebuttal · Authors · 2025-04-01
>
> We thank *reviewer ppW1*’s time and effort for reviewing the manuscript and their constructive comments. Below, we respond to the three concerns raised by the reviewer:
>
> **1. [Average Ranking]**
>
> We respectfully offer a different perspective on this comment. First, it seems that one of our key contributions may have been overlooked: to the best of our knowledge, **LLM-BP is the first approach to design zero-shot graph algorithms that generalize effectively across both homophilic and heterophilic graphs.** We emphasize that strong zero-shot performance on heterophilic graphs is equally important, particularly since prior works have not demonstrated this capability.
>
> To further address the reviewer’s concern, we provide detailed average rankings (based on accuracy and F1 score) across three sub-categories: **Citation** and **E-Commerce** (homophilic), and **School Webpage** (heterophilic). LLM-BP consistently achieves the highest ranking across all three. For clarity and fairness, we will include these sub-category rankings in the revised version of the manuscript.
>
> |  | Citation Graph |  | E-Commerce & KG |  | School Webpage |  |
> |:---:|:---:|:---:|:---:|:---:|:---:|:---:|
> |  | Acc | F1 | Acc | F1 | Acc | F1 |
> | Sbert | 8.3 | 7.3 | 9.3 | 8.8 | 5.8 | 6.0 |
> | Roberta | 7.3 | 7.0 | 7.5 | 8.0 | 6.8 | 7.0 |
> | Text-Embedding-3-Large | 5.3 | 4.3 | 7.3 | 6.3 | 3.3 | 2.5 |
> | LLM2Vec | 8.3 | 5.3 | 6.5 | 5.5 | 3.8 | 4.0 |
> | SBert + NA | 5.0 | 5.0 | 6.0 | 5.8 | 8.3 | 8.3 |
> | GPT-3.5-turbo | 5.0 | 11.3 | 3.5 | 6.3 | 7.3 | 8.3 |
> | GPT-4o | 6.0 | 10.0 | 3.5 | 3.5 | 7.0 | 5.5 |
> | UniGLM | 11.0 | 10.3 | 10.8 | 10.5 | 12.0 | 10.8 |
> | ZeroG | 9.3 | 9.3 | 12.3 | 12.0 | 13.8 | 15.8 |
> | DGI | 15.3 | 15.3 | 15.3 | 16.0 | 15.0 | 15.0 |
> | GraphMAE | 15.7 | 15.7 | 14.3 | 15.0 | 12.3 | 13.8 |
> | OFA | 13.7 | 13.3 | 14.8 | 15.0 | 15.3 | 16.3 |
> | GOFA | 5.7 | 3.7 | 7.0 | 7.3 | 10.3 | 10.5 |
> | GraphGPT | 14.3 | 15.7 | 14.5 | 14.3 | 14.8 | 14.3 |
> | LLAGA | 16.0 | 15.0 | 15.8 | 14.5 | 14.5 | 11.8 |
> | LLM-BP | 3.0 | 2.0 | 2.3 | 3.0 | 1.3 | 1.3 |
> | LLM-BP (appr.) | 3.3 | 2.3 | 2.8 | 1.5 | 1.8 | 2.3 |
>
> **2. [Empirical gain of LLM-BP over GPT-4o on homophily data]**
>
> We respectfully disagree with this comment. As noted in our paper, strong zero-shot performance on heterophilic graphs is equally important, particularly because prior works have not demonstrated such capability. Even on homophilic graphs, LLM-BP still significantly outperforms GPT-4o on 5 out of 7 datasets.
>
> More importantly, LLM-BP introduces a generalizable graph information aggregation mechanism, which GPT-4o fundamentally lacks. This is a core contribution of our work that goes beyond empirical results. Specifically, GPT-4o relies heavily on high-quality node text and cannot leverage graph structure. In contrast, LLM-BP is designed to incorporate structures, which is critical in many real-world scenarios where node text may be noisy or sparse.
>
> To highlight this point, we conducted follow-up experiments on three large citation datasets using the same graph structures but with degraded text inputs: only paper titles were used as node attributes. Under this low-text-quality setting, LLM-BP and its variants significantly outperform GPT-4o, underscoring the value of structural information and the effectiveness of our approach.
>
> |  | Cora |  | Citeseer |  | Pubmed |   |
> |:---:|:---:|:---:|:---:|:---:|:---:|:---:|
> |  | Acc | F1 | Acc | F1 | Acc | F1  |
> | GPT-4o | 58.67 | 48.01 | 62.19 | 46.76 | 70.32 | 51.38  |
> | LLM-BP | 68.91 | 67.36 | 69.04 | 65.98 | 74.36 | 74.06  |
> | LLM-BP (appr.) | 67.5 | 65.62 | 67.87 | 64.92 | 76.68 | 75.55 |
>
>
> **3. [Pre-training Free Claim]**
>
> Thank you for pointing this out. To ensure rigor and avoid any misunderstanding, we will revise the claim in our manuscript to clarify that *“no additional fine-tuning of LLMs is required compared to existing baselines.”*
>
> Our main argument is that LLM-BP does not require further fine-tuning, which leads to significantly improved computational efficiency relative to most baselines that rely on fine-tuning, even if they also leverage LLMs.
>
> -------------
>
> We hope this response addresses the reviewer’s concerns, and we would be happy to provide further clarification if needed.

---

### Official Review · Reviewer_JW3s · 2025-03-20

**Overall Recommendation:** 4

**Summary:**

This paper explores zero-shot generalization in graph problems on Text-Attributed Graphs (TAGs) using a pure LLM-based approach. The authors propose two key principles for model design:

- Task-Adaptive Embeddings – An LLM-based encoder processes raw node text along with a prompt, allowing node embeddings to dynamically adjust based on the prompt content.
- Graph Aggregation System – The graph is modeled as a Markov Random Field (MRF), and Belief Propagation (BP) is mimicked to perform aggregation.

The proposed approach is evaluated on graph datasets from multiple domains, considering both homophilic and heterophilic scenarios.

**Claims And Evidence:**

The claims are accurate and supported by clear evidence.

**Essential References Not Discussed:**

The authors made a comprehensive discussion of the related works.

**Experimental Designs Or Analyses:**

N/A

**Methods And Evaluation Criteria:**

The model design is promising, built on two important principles. The idea of using prompts along with node text as input to an LLM encoder to learn adaptive node embeddings is particularly interesting.

However, I have a question regarding the second principle. In the LLM-BP algorithm, the method for calculating class embeddings seems to rely on the assumption that there is abundant data for each novel class. Is that correct? If a category is truly new and has very limited data, how would this method adapt? What is the reasoning behind this design choice?

The evaluation is also well-structured, including zero-shot baselines from different model types (LLM, GNN, LLM+GNN) and demonstrating performance improvements on these tasks. Additionally, the visualization of the embedding space strengthens the claims made in the paper.

**Other Comments Or Suggestions:**

N/A

**Other Strengths And Weaknesses:**

- Strength:
1) The paper is well-written, clearly presenting both the motivation and model details.
2) The model design is interesting and thoughtfully structured.
3) The experiments demonstrate strong performance, supporting the proposed approach.
- Weakness:
1) Please refer to the Methods and Evaluation Criteria section for specific concerns.
2) Additionally, I wonder whether this approach can be generalized to zero-shot graph classification tasks. Another potential limitation is that the model output is still not fully flexible, which makes it difficult to handle zero-shot QA tasks effectively.

**Questions For Authors:**

- I notice the link prediction zero-shot performance is pretty high compared to baselines, could you explain the possible reason?

**Relation To Broader Scientific Literature:**

N/A

**Theoretical Claims:**

N/A

---

> ### Author Rebuttal · Authors · 2025-04-01
>
> We sincerely thank *reviewer JW3s*’s time and effort in reviewing the paper. We are also thankful for the constructive suggestions. Below we try to address the concerns from *reviewer JW3s*:
>
> **1. [New Class with limited data]**
>
> LLM-BP does not require abundant data from new classes and can generalize to novel classes even when data is limited. Consider an extreme cold-start scenario with only a single node from a new class:
>
> Due to the scarcity of nodes in the new class, it is unlikely that edge probability estimation will involve edges connected to this new node. As a result, the estimated edge probabilities between the new class and known classes will be extremely low. According to the BP rule, this leads to an aggregation that minimizes the influence of neighboring nodes on the new node’s label. This behavior aligns well with standard practice in cold-start settings, where predictions for new classes should rely more on the node’s own attributes than on potentially misleading signals from sparse or noisy connections.
>
> Regarding class embeddings, one could directly use text descriptions of the classes. However, we chose not to adopt this approach in the current work, as the resulting embeddings can vary depending on the phrasing of the description. Instead, we propose a more robust strategy: sampling a few nodes from each class and aggregating their embeddings to form a stable class-level representation. That said, for cold-start scenarios, using text-description-based class embeddings remains a viable and potentially effective alternative.
>
> **2. [Applying LLM-BP to Graph-level Tasks]**
>
> This issue is precisely the focus of our planned future research. The current LLM-BP algorithm cannot be directly applied to graph-level tasks, as both the task-adaptive embedding and the BP mechanism are specifically designed for node-level settings.
>
> That said, this limitation does not undermine the core message of our work: when resources are insufficient to train complex adaptors that align graph data with LLMs, strong generalization requires algorithmic designs that (1) adhere to two high-level principles (attribute unification and unified information aggregation) and (2) are tailored to the structure of the downstream task. In this work, we focus on node-level tasks as a proof of concept to illustrate these principles, demonstrating that they can lead to strong generalization performance even under resource constraints.
>
> **3. [Limitation in QA tasks]**
>
> We agree with the reviewer’s point. Flexible QA should further involve LLM decoders, while how to combine graph structure with LLM decoders in a practically generalizable way remains an unsolved question and we are eager to research with.
>
> **4. [zero-shot link prediction]**
>
> For zero-shot link prediction tasks, our explanation is as follows: For the baseline, as mentioned in the paper, the LLM-with-adaptor baselines are not trained on link prediction tasks and are only trained on node-level tasks, while the strong performance of ours comes from the high-quality node embedding via BP.

---

### Decision · Program_Chairs · 2025-05-01

**Decision:**

Accept (poster)

**Comment:**

The paper proposes using an LLM-based approach to enhance graph learning on text-attributed graphs. The method is novel, and extensive evaluations are conducted on graph datasets from various domains, demonstrating significant performance improvements over baseline methods. Overall, this is a solid work. To improve the final paper, the authors should clarify the utility of graph structure, provide performance details on both homophilic and heterophilic datasets, and explain incremental improvements specifically on homophilic datasets.